# Recent Advances in the Synthesis of Complex Macromolecular Architectures Based on Poly(N-vinyl pyrrolidone) and the RAFT Polymerization Technique

**DOI:** 10.3390/polym14040701

**Published:** 2022-02-11

**Authors:** Nikoletta Roka, Olga Kokkorogianni, Philippos Kontoes-Georgoudakis, Ioannis Choinopoulos, Marinos Pitsikalis

**Affiliations:** Industrial Chemistry Laboratory, Department of Chemistry, National and Kapodistrian University of Athens, Panepistimiopolis Zografou, 15771 Athens, Greece; nikolettaroka@yahoo.gr (N.R.); olga.kokkorogianni@gmail.com (O.K.); kontoes10@hotmail.com (P.K.-G.); ichoinop@chem.uoa.gr (I.C.)

**Keywords:** poly(N-vinyl pyrrolidone), RAFT polymerization, statistical copolymers, block copolymers, star polymers, graft copolymers, macromolecular architecture

## Abstract

Recent advances in the controlled RAFT polymerization of complex macromolecular architectures based on poly(N-vinyl pyrrolidone), PNVP, are summarized in this review article. Special interest is given to the synthesis of statistical copolymers, block copolymers, and star polymers and copolymers, along with graft copolymers and more complex architectures. In all cases, PNVP is produced via RAFT techniques, whereas other polymerization methods can be employed in combination with RAFT to provide the desired final products. The advantages and limitations of the synthetic methodologies are discussed in detail.

## 1. Introduction

In 1938, one year before the declaration of World War II, Walter Reppe synthesized N-vinyl pyrrolidone (NVP) from acetylene and formaldehyde. In 1939, the radical polymerization of this compound produced a material which was used in World War II as a plasma substituent, due to the fact that it is hemocompatible and physiologically inactive. This polymeric material, poly(N-vinyl pyrrolidone) (PNVP), has found many biomedical and other applications since then [1].

Amongst the various properties of PNVP, biocompatibility, lack of toxicity, water solubility and the complexation of hydrophobic compounds are considered crucial for biomedical applications. Other properties, such as film formation, chemical and thermal resistance, as well as its amorphous nature, are also exploited in the production of pharmaceuticals and other products [2].

The versatile properties of PNVP have led to applications in a variety of fields:

Pharmaceutical and biomedical: binder, coating and stabilizer for tablets, solubilizer for suspensions, disinfectant solutions (PNVP-Iodine), dispersion of crystallizing drugs, hydrogels/nanogels, nanocarriers for drug/gene/protein/peptide delivery, tissue engineering (scaffolds), wound/burn dressings, dental restoration, wettability of contact lenses, and antifouling agents [1,2,3,4,5,6,7,8,9,10,11,12].

(a)Laboratory analysis: protein and DNA analysis, as sieving or shielding medium and pollutant analysis as sorbent [12,13,14].(b)Cosmetics: film for hair dressing products, setting lotions and conditioning shampoos [15,16,17].(c)Food products: stabilization of beverages (polyphenol remover) [8,16,17].(d)Adhesive sticks and remoistenable adhesives [16,17].(e)Suspending agent in two-phase polymerization systems [17].(f)Dye-affinitive stripping and levelling agent in textile processing [16,17,18](g)Fuel cells and batteries [19,20](h)Metal nanoparticle synthesis: surface stabilizer, growth modifier, nanoparticle dispersant, and reducing agent, shape-control of metal nanowires, nanospheres, nanoplates and nanobelts [21,22].(i)Environmental protection: removal of heavy metals [15].

Apart from linear copolymers, well-characterized advanced polymeric structures cannot be synthesized by free radical polymerization [23]. The discovery of methods which control radical polymerization has provided the means to synthesize well-characterized complex macromolecular architectures. One of these methods is Reversible Addition-Fragmentation chain Transfer (RAFT) polymerization mediated by thiocarbonylthio chain transfer agent (CTA) compounds and was invented in 1998 [24]. Since then, a plethora of CTAs has been used in the controlled radical polymerization of a wide variety of monomers.

In comparison with their linear homopolymers, complex macromolecular architectures such as statistical, block, graft, comb, and star copolymers, may lead to amphiphilic, amorphous–crystalline, flexible–rigid and other advanced structures [23]. These structures present unique properties, including phase separation, micellization behavior, thermal properties, formation of nanoparticles, etc. The exploitation of the properties of these materials leads to new applications. Thus, the synthesis and characterization of complex macromolecular architectures is of great significance. Obviously, there are infinite combinations of monomers and architectures. The selection of the correct combination is crucial in order to achieve the desired properties. Since there are no rules for the correct combination, experience obtained by studying these materials is very important.

In this review, we aim to present the synthesis of macromolecular architectures consisting of or containing segments of poly(N-vinyl pyrrolidone) (PVP) synthesized using Reversible Addition-Fragmentation chain Transfer (RAFT) polymerization. NVP can be polymerized only through radical polymerization. Among the various controlled radical polymerization techniques, such as Atom Transfer (ATRP) and Nitroxide-Mediated (NMP) Radical Polymerization, RAFT has been proven to be the only methodology available to provide control over the molecular characteristics of PNVP and the possibility to synthesize complex structures based on PNVP components. In addition, RAFT can be combined with other controlled/living polymerization techniques, thus providing the possibility to synthesize structures with unique properties and possible interesting applications.

## 2. Principles of RAFT Polymerization Technique

Reversible Addition-Fragmentation chain Transfer (RAFT) polymerization, Atom Transfer Radical Polymerization (ATRP) and Nitroxide-Mediated Polymerization (NMP) are the three most common Reversible-Deactivation Radical Polymerization (RDRP) processes. Their most important feature is the ability to control the radical polymerization process in a way that the composition, the architecture and the polydispersity can be tuned at will [25].

A simplified mechanism of activation–deactivation equilibria in RAFT polymerization is presented in Figure 1.

The propagating radical species can covalently bond to a thiocarbonylthio group to form the intermediate radical species. These species further fragment at the C-S bond to form a dormant polymeric chain and a new propagating radical. This process can be proven successful if the intermediate radical species fragment rapidly. Ιn this case, all polymeric chains react at approximately the same rate. Thus, narrow molecular weight distributions are obtained. When the polymerization is terminated, most chains retain their thiocarbonylthio end-groups, which allows repetition of the propagation step with the same or other suitable monomers.

RAFT polymerization takes place via employment of a Chain Transfer Agent (CTA) (Figure 2). A wide variety of thiocabonylthio compounds, such as xanthates, trithiocarbonates, dithiocarbamates, etc., has been used as CTAs.

The effectiveness of the CTA is determined by the properties of the R and Z substituents. Both the nature of the monomer and the nature of the CTA influence the polymerization procedure. Thus, careful selection of CTA and monomer has to be made in order to obtain the appropriate control over the polymerization reaction. Monomers are categorized as ′More-Activated Monomers′ (MAMs) and ′Less-Activated Monomers′ (LAMs). N-vinyl pyrrolidone (NVP) is an LAM and xanthates are the most suitable CTAs, providing good control of the NVP polymerization [26].

The most important advantages of RAFT polymerization include (a) good control of the polymerization of most monomers [(meth)acrylates, (meth)acrylamides, acrylonitrile, styrenes, dienes, and vinyl monomers]) (b) monomer functional group tolerance (-OH, -NR_2_, -CONR_2_, -COOH, -SO_3_H, etc.), and (c) use of organic solvents, as well as aqueous media. The absence of metals in RAFT polymerizations is a definite advantage in comparison with ATRP. Furthermore, the high versatility of RAFT experimental conditions is significantly beneficial compared to those of NMP.

RAFT can produce polymers exhibiting a molecular weight under 1,000,000 g/mol with excellent control compared to the stoichiometric values. Polymers with molecular weight over 1,000,000 g/mol can also be synthesized but require more specific reaction conditions based on kinetic parameters [27].

The retention of the thiocarbonylthio group at the end of the polymeric chains allows for the synthesis of diblock and higher order block copolymers. Furthermore, RAFT agents of appropriate design are able to provide more complex macromolecular architectures, e.g., star polymers, graft copolymers, etc.

As shown in Figure 3, a conventional radical initiator is used. However, in the presence of the CTA (Figure 3, (1′)) the polymerization does not proceed through the radicals formed by the initiator but rather from the initiating radicals P_m_. These radicals are produced from R**^.^** (Figure 3, (4′)), which is the fragmentation product of the intermediate (2′). This is achieved by utilizing a low concentration of initiator relative to CTA and a much higher reactivity of the CTA compared to the monomer. Equation (1’) (II) including the consumption of CTA and reversible fragmentation of species (2) is usually referred to as pre-equilibrium in order to differentiate from Equation (4’) (IV), which is the main equilibrium. The most important requirements for producing polymers with controlled molecular weights and narrow molecular weight distributions are the following: (a) rapid establishment of the pre-equilibrium, (b) efficient re-initiation by the R**^.^** fragment and (c) attainment of the main equilibrium in which the population of dormant chains and/or intermediate radicals (Figure 3, (5′)) (not reactive enough to add to monomers) is much higher than the total number of propagating chains P_n_ and P_m_. RAFT is an extremely versatile method regarding the monomer functionality and rigorous experimental techniques (vacuum line, use of extra pure reagents, etc.) are not required.

## 3. Polymerization of NVP by RAFT

RAFT can be applied to a huge variety of monomers, which are susceptible to radical polymerization. They can be classified into two categories, the more-activated (MAMs) and the less-activated monomers (LAMs), according to their ability to react with free radicals [27]. MAMs form more-stabilized and less reactive radicals than LAMs due to electronic stabilization and even steric effects. MAMs exhibit a double bond conjugated to another double bond, an aromatic ring, a nitrile or a carbonyl group, whereas LAMs have their double bond connected to saturated carbons or adjacent to nitrogen, oxygen, halogens, etc. Typical examples of MAMs include dienes (butadiene, isoprene), acrylonitrile, (meth)acrylates, styrene, vinyl pyridine, and (meth)acrylamides, whereas typical examples of LAMs are vinyl alkanoates (e.g., vinyl acetate, vinyl butyrate, vinyl stearate), vinyl carbazole, vinyl chloride and 1-alkenes.

The monomer of interest of this work, N-vinyl pyrrolidone (NVP), belongs to the family of LAMs. Different classes of RAFT agents have to be employed to promote the controlled polymerization of MAMs and LAMs. The role of the Z and R groups of the CTA are crucial for the successful transfer reaction and the control of the molecular characteristics of the polymeric products. The crucial parameter is that the CTA’s C=S bond should be more reactive to radical addition than the monomer’s double bond. MAMs lead to the formation of more stabilized radicals due to the electronic stabilization, which is affected by the substituents conjugated to their double bond. Consequently, these monomers require a Z-group able to stabilize the intermediate radical, thus promoting radical addition on the C=S group. Trithiocarbonates and dithiobenzoates are among the most successful CTAs for the well-controlled polymerization of MAMs, since they provide a high rate of reversible chain transfer via addition-fragmentation over the propagation. On the other hand, O-alkyl xanthates or N-alkyl-N-aryldithiocarbamates, which form less-stable intermediate radicals, are more appropriate CTAs for the RAFT polymerization of LAMs, since these monomers are highly reactive and behave as poor homolytic groups.

Consequently, the CTAs that have been reported for the controlled polymerization of NVP are several xanthates and dithiocarbamates, as given in Figure 1.

Before the appearance of the RAFT approach, the polymerization of NVP was conducted mainly by conventional radical polymerization [29,30]. Even though the polymerization could be performed following an easy protocol, the molecular characteristics of the resulting polymers were far from controlled. On the contrary, the RAFT technique combined with the use of the proper CTA, as described above, leads to polymers of PNVP with a relatively narrow molecular weight distribution and controlled molecular weights, sufficient for most applications. It is noteworthy that over the last several years, the polymerization of the NVP has been conducted almost entirely via RAFT.

NVP polymerization, like all RAFT polymerizations, exhibits to a small extent nonideal kinetic behavior, such as induction period and rate retardation [31]. The induction period is a retarding kinetic effect during which there is no polymerization activity in the initial phase of the polymerization. The rate retardation is the effect during which the polymerization is slower in rate, in comparison with the corresponding conventional radical polymerization. In addition, the impurities of the RAFT agent may cause side reactions producing several by-products. These effects can be almost eliminated by the right choice of the RAFT agent as well as the use of the proper amount of both the RAFT agent and the initiator [32,33].

A successful RAFT polymerization leads to polymer chains with α and ω end-groups incorporated. The two end-groups are, respectively, the R group and the thiocarbonylthio functional Z group of the RAFT agent [34]. The thiocarbonylthio group can be efficiently removed if this is required for the subsequent use of the polymer. In contrast, for some other applications, the presence of this group is a prerequisite, as it can provide the polymer with distinct properties. In particular, it can be used as a masked thiol and at the desired time, a reduction of thiol can take place by either hydrolysis or aminolysis. This process is one of the most well-known and applied techniques, but the fact that the Z groups are quite unstable to nucleophiles cannot be underestimated. For this reason, the in situ synthesis of end-functional polymers is traditionally conducted by using the R-group of the RAFT agent [35].

Ultimately, the RAFT method has the capability to lead to versatile end-functionalized PNVP polymers by simply choosing the proper CTA, or even better, by using a tailor-made CTA. Interestingly, in several cases, the end-functionalized PNVP polymers can be post-modified to provide other useful end-functional groups [36].

## 4. Synthesis of Complex Macromolecular Architectures

### 4.1. Statistical Copolymers

#### 4.1.1. Introduction

Statistical copolymers are a group of very fascinating copolymers, which can be further classified as random, gradient, and alternating copolymers according to the sequence of the monomeric units along the macromolecular chain. A large number of statistical copolymers has been synthesized via RAFT in an easy, one-step procedure from a mixture of A and B monomers in the presence of the suitable RAFT agent and radical initiator. In contrast, the synthesis of block copolymers and other complex macromolecular architectures requires multistep reaction pathways [37]. The copolymers are usually synthesized in order to attain or improve certain properties. The diverse succession of chemical bonds mostly results in unexpected copolymer properties, which of course are not the arithmetic average or linear variation of the corresponding homopolymers. The copolymer properties are determined by the chemical structure, the composition, and the sequence of the monomeric units, as well as by the molecular weight of the macromolecular chain. In addition to all these parameters, statistical copolymerization remains one of the most attractive and easy procedures to afford tailor-made polymers with specific and desired properties [38].

#### 4.1.2. Evaluation of the Reactivity Ratios

A significant issue in statistical copolymerization is the employment of a suitable method for the evaluation of the reactivity ratios of the monomers involved. When the reactivity ratios are identified for a set of monomers, the structural characteristics of the copolymers, as well as the polymerization rate, can be theoretically predicted.

Traditionally, the binary copolymerization parameters are obtained by the terminal model, applying linearized versions of the differential form of the Mayo–Lewis copolymerization equation to the obtained experimental data, and fitting straight lines by graphical or numeric regression analysis [38]. According to this model, the chemical reactivity of the propagating chain depends only on the monomer unit at the growing end. Well-known linear least square methods for the estimation of the reactivity ratios by terminal model include the Fineman–Ross [39], inverted Fineman–Ross [39], Kelen–Tüdos [40] and extended Kelen–Tüdos methods [40].

When the fitting of the linearized equations deviates from linearity, or when the copolymerization parameters reveal an apparent negative value, it can be concluded that the terminal model is a poor choice for the system, and that the penultimate model should be investigated. The penultimate model is based on the Merz–Barb–Ham equation, and the chemical reactivity of the propagating chain depends not only on the monomer unit at the growing end but also on the nature of the monomer unit preceding the last one. The penultimate effect is expected to be applied when at least one of the comonomers is bulky and introduces extended steric effects when it is polar or has strong electron-withdrawing groups and displays resonance effects [41,42]. According to this method, each monomer is characterized by two monomer reactivity ratios, one representing the propagating species in which the penultimate and the terminal monomer units are the same and the other representing the propagating species in which the penultimate and terminal units differ. A well-known graphical method for the estimation of the reactivity ratios with the penultimate model is the linearized Barson–Fenn method [43].

Furthermore, there are some rarely used models for the description of copolymerization, such as the depropagation model based on the Lowry equations [44], and the complex participation model based on the Seiner–Litt equation [45].

All the linear least square methods mentioned above have statistical limitations inherent to the applied linearization, since the independent variable of the linear equations is not really independent and since the variance of the dependent variable is not constant. Almost all these approaches were revolutionized by the use of computer programs. Among them, the COPOINT program [46] that fits integrated copolymerization equations to experimental monomer/copolymer composition data by means of non-linear least square difference procedures has received significant interest. The program applies numeric integration techniques in their differential form. The copolymerization parameters can be obtained by minimizing the sum of square differences between measured and calculated polymer compositions. In addition, the user can select between several copolymerization equations, such as that of the terminal model or that of the penultimate model. Several other non-linear regression methods and computer programs have been applied for the more accurate determination of the monomer reactivity ratios, including the Tidwell–Mortimer [47] and the errors-in-variables model [48].

Nevertheless, the nature of the copolymers, as well as their properties, depend not only on the monomeric unit composition described by the reactivity ratios, but also on the arrangement of the various monomeric units along the polymer chain backbone. Ultimately, the copolymer structure is described by the statistical distribution of the dyad and triad or even higher monomer sequences and can be calculated using the corresponding equations proposed in the literature, such as the Igarashi equations [49] or can be calculated by NMR techniques [38,50]. In addition, there are equations for the estimation of the mean sequence lengths as the reciprocal of the conditional probability [37].

#### 4.1.3. Statistical Copolymers of NVP via RAFT

In the literature, there are two groups of works related to statistical copolymers. On the one hand, there are works in which the central issue is the statistical copolymers themselves: their synthesis, their characterization and any other information confirming the value of these copolymers. On the other hand, there are more complicated works in which the statistical copolymers are employed as scaffolds for the synthesis of more complex macromolecular architectures.

In a series of publications, the synthesis of statistical copolymers of NVP with various methacrylates was presented in an effort to trace the best copolymerization model, which efficiently describes the copolymerization process, to calculate the reactivity ratios of the comonomers and to study the structural characteristics of the copolymers. The most recent project described the synthesis of statistical copolymers of NVP with isobornyl methacrylate, IBMA [51], which was conducted at 60 °C by using AIBN as initiator, [(O-ethylxanthyl)methyl]benzene as CTA and 1,4-dioxane as solvent. The polymers were precipitated in cold methanol. The reactivity ratios were estimated using the Barson–Fenn equation and the computer program COPOINT tuned to the penultimate model. The efforts to employ the terminal model for this system, applying almost all the available methods for the determination of the reactivity ratios, proved fruitless, since in all cases the reactivity ratio for NVP was negative. The huge difference in polarity between the two monomers and the bulkiness of IBMA, compared to NVP, were decisive parameters leading to the application of the penultimate model. Finally, the NVP reactivity ratio was significantly lower than that of IBMA, implying a tendency for pseudo- or gradient diblock synthesis. The thermal properties of the copolymers were studied by differential scanning calorimetry (DSC), thermogravimetric analysis (TGA), and differential thermogravimetry (DTG). The activation energies (Ea) were calculated by the Kissinger and Ozawa–Flynn–Wall (OFW) methodologies, concluding that the statistical copolymers showed a similar multistep complex thermal degradation mechanism.

The statistical copolymerization of NVP with the alkyl methacrylates, hexyl methacrylate, HMA, and stearyl methacrylate, SMA, was also described [52]. The synthesis of the copolymers and the estimation of the reactivity ratios were initially utilized, followed by the study of the kinetics of the thermal degradation of these copolymers [53]. The synthesis of the copolymers was conducted employing [(O-ethylxanthyl) methyl]benzene and [1-(O-ethylxanthyl) ethyl]benzene as the RAFT agents and AIBN as initiator. The copolymerization reactions of HMA were performed in bulk, whereas the copolymerization reactions of SMA in THF. All the obtained polymers were precipitated in cold methanol. Subsequently, the reactivity ratios were estimated using classic graphical methods as well as the computer program COPOINT, modified to both terminal and penultimate models. As in the case of the statistical copolymers with IBMA, the penultimate model fits the experimental results better than the terminal model. In all cases, the NVP reactivity ratio is much lower than that of the methacrylates, implying a tendency for pseudo- or gradient diblock synthesis. The thermal properties of the copolymers were studied by DSC and TGA, and the results were compared to those of the respective homopolymers. In spite of the relatively small amount of NVP in the copolymers, their thermal properties were influenced by both components. Moreover, the activation energies of the thermal decomposition were calculated using the Ozawa–Flynn–Wall, the Kissinger, and the Kissinger–Akahira–Sunose methodologies. It was found that the presence of NVP units considerably increases the activation energy values, which are relatively close to those obtained from the PNVP homopolymer.

The statistical copolymerization of NVP with benzyl methacrylate, BzMA, has also been reported [54]. The copolymerization was conducted in bulk at 60 °C, by employing three different xanthates as RAFT agents: [(O-ethylxanthyl)methyl] benzene, [1-(O-ethylxanthyl)ethyl] benzene and O-ethyl S-(phthalimidylmethyl) xanthate. AIBN was the initiator, and the produced polymers were precipitated in cold methanol. The reactivity ratios were measured in the framework of several graphical and non-graphical methods, including COPOINT. All methods indicated that the BzMA reactivity ratios were much larger than those of NVP. The *T*g values of the copolymers were measured by DSC, while a systematic and detailed investigation of the thermal degradation was carried out by TGA, leading to the conclusion that the thermal stability of the copolymers was influenced by the structure of the substituents of the thiocarbonylthio end groups due to the RAFT agents.

The synthesis of statistical copolymers of NVP with 2-(dimethylamino)ethyl methacrylate, DMAEMA, was utilized, in bulk, at 60 °C by employing three different xanthates as RAFT agents: [(O-ethylxanthyl)methyl] benzene, [1-(O-ethylxanthyl)ethyl] benzene and O-ethyl S-(phthalimidylmethyl) xanthate and AIBN as the initiator [55]. The copolymers were precipitated in n-hexane. The reactivity ratios were estimated mainly by the computer program COPOINT, revealing that the DMAEMA reactivity ratio is much greater than that of NVP; thus, the statistical copolymers are in fact pseudo-diblocks. Similar results were obtained with all three CTAs, indicating that their nature does not noticeably affect the copolymerization behavior, since they have more or less similar chemical structures. The *T*g values of the copolymers were measured by DSC, and it is worth noting that for the copolymers synthesized by CTA 1, three distinct *T*g values were found for every copolymer, implying microphase separation. Subsequently, the thermal degradation of the copolymers was analyzed by thermogravimetric analysis, TGA, and differential thermogravimetry, DTG, using the Ozawa–Flynn–Wall and Kissinger methodologies, showing that the thermal stability of the copolymers is influenced by both monomers and by the structure of the thiocarbonylthio end groups due to the RAFT agents.

Forced-gradient (block-like polymers, BLG) and block copolymers of NVP with vinyl laurate (VL) were prepared [56] in order to investigate the effect of polymer chain architecture on the aqueous solution properties. The statistical copolymers were produced using S-(2-ethyl propionate)-O-ethyl xanthate as RAFT agent, AIBN as initiator, 1,4-dioxane as solvent and anisole as an internal standard for ^1^H NMR analysis, whilst the products were precipitated in diethyl ether. The BLGs were produced using the same materials, changing only the order in which they were added. VL, the less reactive monomer, was polymerized first until the conversion reached ~50%, when the second monomer, NVP, was added and allowed to copolymerize with the remaining amount of VL. The monomer reactivity ratios of the statistical copolymers were estimated by the Kelen–Tüdos method, indicating that the ratio of NVP is greater than the one of VL. In all cases, copolymers of relatively high molecular weight and very broad molecular weight distribution (Đ > 2 for all samples) were produced. After the dynamic light scatting analysis of aqueous solutions, the final conclusion was that the BLGs more easily behave in a similar fashion to that of the block copolymers, forming micelles and exhibiting lower critical micelle concentration, CMC, values compared to the statistical copolymers.

An additional set of statistical copolymers was reported from the combination of NVP with the functional methacrylamide monomer N-[2-(3-hydroxy-2-methyl-4-oxopyridin-1(4H)yl)ethyl]-2-methylprop-2-enamide [57]. The copolymerization was conducted at 40 °C in aqueous solutions under mild conditions, with 2-[(ethoxymethanethioyl) sulfanyl]-2-methylpropanoic acid as the RAFT agent, in the presence of N,N,N′,N′-tetramethylethylenediamine and tert-butyl hydroperoxide, followed by precipitation in methyl t-butyl ether. The samples were characterized by SEC and NMR spectroscopy. The good correlation between refractive index and UV-vis profiles indicated a homogeneous distribution of the functional monomer across the polymeric chain. It is noteworthy that the produced copolymer could not incorporate amounts higher than 15 mol% of the functional monomer. Samples of low molecular weight (less than 5000) and moderate dispersity values (1.4 < Đ < 1.9) were obtained. Therefore, a novel family of extremely water-soluble PNVP-based copolymers was developed to selectively bind iron(III) ions and to behave as an antiseptic.

The synthesis of statistical copolymers of NVP with 3-ethyl-1-vinyl-2-pyrrolidone, C2NVP, was reported [58]. The copolymerization was conducted at 70 °C in ethanol solutions employing S-(1-methyl-4-hydroxyethyl acetate) O-ethyl xanthate, a difunctional CTA, and AIBN as initiator. Finally, the copolymers were precipitated in n-hexane. Thus, a series of well-defined copolymers with relatively low Đ values (Đ < 1.5) was synthesized, presenting characteristic lower critical solution temperature (LCST) behavior. The reactivity ratios were estimated by the Kelen–Tüdos method, revealing a similar reactivity ratio for the two monomers and predominantly ideal random copolymerization, while the kinetic study also gave similar reaction rates for the two monomers. In addition, the study of the cloud points (CPs) of the copolymers was performed. Chain extension of the copolymers was subsequently achieved with the combination of ring-opening polymerization of ε-caprolactone, leading to the amphiphilic block copolymer P(C2NVP-co-NVP)-b-PCL.

The synthesis of the statistical copolymers of NVP with N-vinyl formamide, VFA, was achieved using methyl 2-(ethoxycarbonothioylthio)propanoate as RAFT agent, AIBN as initiator, anisole as a solvent at 60 °C, as well as 1,3,5-trioxane as internal standard for monomer conversion calculation via NMR spectroscopy [59]. The copolymers were precipitated in diethyl ether followed by a systematical characterization by SEC, NMR and FT-IR/Raman spectroscopies. Copolymers of low molecular weight (less than 10,000) and relatively low Đ values (1.13 < Đ < 1.46) were produced. Additionally, the controlled hydrolysis of the VFA components under alkaline conditions led to primary amine functionalized, temperature/pH dual responsive reactive copolymers. Subsequently, amine-enriched polymeric nanogels were prepared by direct crosslinking of the amine groups and vinyl groups via the typical Michael addition reaction in water-in-oil emulsion, followed by labelling with fluorescein isothiocyanate (FITC) through conjugation with the residual primary amine groups on the surface. This facile chemistry has been applied towards the synthesis of water-soluble reactive copolymers with well-defined architectures for fabrication of redox-sensitive degradable prodrug nanogels for intracellular drug release [60].

Statistical copolymers of NVP with N-(methacryloxy)succinimide were synthesized via RAFT [61]. The reaction was carried out at 80 °C in anisole solutions involving methyl 2-(ethoxycarbonothioylthio) propanoate as RAFT agent and AIBN as initiator. The copolymers were precipitated in diethyl ether and characterized by SEC, NMR and FT-IR/Raman spectroscopies. The produced water-soluble copolymers were directly combined with enhanced green fluorescent protein (EGFP) or cellulase (CelA2_M2) at room temperature in a water-in-oil emulsion in order to synthesize biohybrid nanogels. The EGFP-conjugated nanogels were fluorescent, while the CelA2_M2, which was encapsulated in the nanogels, demonstrated relatively high catalytic activity. In a second study of this approach, more detailed analysis revealed that the biocatalytic activity of cellulase-conjugated nanogels (CNG) can be elegantly tuned by control of their crosslinking densities [62].

Statistical copolymers of NVP with pyridyl disulfide ethyl methacrylate were successfully prepared via RAFT [63]. The copolymerization reaction was carried out at 60 °C, in anisole solutions using methyl 2-(ethoxycarbonothioylthio)propanoate as RAFT agent, AIBN as initiator, as well as 1,3,5-trioxane as internal standard for monitoring the monomer conversion via NMR spectroscopy. The copolymers were precipitated in diethyl ether and characterized by SEC, NMR and FT-IR/Raman spectroscopies. Copolymers of low molecular weights (less than 8000) and relatively low dispersities (Đ < 1.30) were obtained. Subsequently, the pyridyl disulfide (PDS)-functionalized reactive polymers were amenable to further functionalization with a variety of thiol-containing molecules, ligands or proteins, via a highly selective thiol−disulfide exchange reaction under mild conditions. The conversions in all cases were higher than 95%, indicating that the thiol−disulfide exchange reaction to PDS groups with thiol-containing molecules is highly selective and tolerant to different ligands, providing a versatile scaffold for facile conjugation of various biological components.

The synthesis of thermoresponsive statistical copolymers of NVP with N-vinylcaprolactam, NVCL, was demonstrated using O-ethyl-S-(1-methoxycarbonyl)-ethyldithiocarbonate as RAFT agent, AIBN as initiator and dioxane as solvent [64]. The copolymerization was conducted at 65 °C. The products are precipitated in diethyl ether and characterized by a variety of techniques, such as SEC in THF, NMR spectroscopy, Dynamic Light Scattering, UV-vis spectroscopy, DSC, Cryo-Scanning Electron Microscopy (Cryo-SEM) and rheological measurements. Several copolymers have been synthesized, with predetermined molecular weights and compositions up to high conversion, in order to investigate the effect of the copolymer composition on their thermoresponsive behavior and hydrogel properties. DSC measurements supported with statistical calculations proposed that the whole polymer chain was involved in the hydration/dehydration process and not only short polymer sequences. Moreover, the copolymers enabled the formation of thermoresponsive hydrogels at high concentration. Cryo-SEM analysis of different systems showed in all cases the presence of globular substructures, with a less-dense network structure at higher NVP content, which could interpret the reduction of the mechanical properties and the faster rehydration kinetics of the thermogelling polymers.

The same statistical copolymers of NVP and NVCL were prepared using different polymerization conditions: [(S)-2-(ethyl propionate)-(O-ethyl xanthate)] was used as RAFT agent, AIBN as initiator and anisole as solvent [65]. The reaction took place at 60 °C. The reactions were performed using an automated parallel synthesizer. After the end of the copolymerization, the polymers were precipitated in hexane. The samples were characterized by SEC and NMR spectroscopy, and the cloud point temperatures were measured. Stable colloidal solutions of gold nanoparticles, AuNPs, coated with the thermoresponsive homopolymer PNVCL and the statistical copolymers of NVP with NVCL, were subsequently obtained via a direct ‘grafting to’ approach. Finally, the hydrodynamic size was measured by DLS, while the polymer coating was visualized by TEM. Thermoresponsive polymer-coated AuNPs have a significant value due to their response to external stimuli, such as temperature, pH and salt.

A third approach for the synthesis of statistical copolymers of NVP with NVCL was carried out at 90 °C using 2-cyano-5-hydroxypentan-2-yl dodecyl trithiocarbonate as CTA, 4,4′-azobis(4-cyanopentanol) (ACP) as initiator and 1,4-dioxane as solvent [66]. The purification was carried out by precipitation in diethyl ether and decantation. Samples of relatively low dispersity values were prepared (Đ < 1.24). It was imperative to keep the NVP content low (lower than 15%) in order to study the effect of the comonomer to the cloud point temperature, *T_CP_*, of the PNVCL homopolymer. The use of this particular RAFT agent was crucial, as it can be used in different families of monomers, both MAMs and LAMs. Subsequently, the polymers were used as macro-CTAs for the preparation of diblock copolymers. The prepared statistical and block copolymers showed a variable cloud point temperature *T_CP_* depending on the comonomer type, the comonomer content and the pH of the aqueous solution. By this synthetic methodology, a copolymer with a targeted *T_CP_* can be prepared.

The synthesis of statistical copolymers of NVP with ribavirin (RBV) acrylate was demonstrated to afford macromolecular prodrugs of RBV [67]. The synthesis was conducted in DMSO solutions at 60 °C with phthalimidomethyl-O-ethyl xanthate as RAFT agent and AIBN as initiator, while the copolymers were precipitated in ether. The samples were characterized by SEC and NMR spectroscopy. Ribavirin (RBV) is a broad-spectrum antiviral agent, as well as a standard medication against hepatitis C virus. Despite decades of clinical success, this therapeutic agent exhibits unfavorable pharmacokinetics, while the resulting polymer therapeutics were effective in delivering their payload to the cultured macrophages and afforded a significantly wider therapeutic window. Furthermore, these statistical polymers have been designed for effective virus inhibition and as antiviral drug delivery carriers. Therefore, these materials have the potential to significantly improve the efficacy of antiviral therapeutics and provide a perspective on polymer-based approaches for the treatment and prevention of coronavirus infection [68].

The equimolar copolymerization of NVP with 1,1,1-3,3,3-hexafluoroisopropyl-α-fluoroacrylate was presented in THF or methyl ethyl ketone, MEK, solutions at 60 °C involving benzyl dithiobenzoate as CTA and AIBN as initiator. The products were precipitated in petroleum ether. According to the reactivity ratios, the system of these two monomers produces highly alternating copolymers. In a first work, the synthesis of these polymers by both RAFT and conventional radical polymerization took place, in order to compare the results of these methods [69]. Subsequently, in a second project, the same copolymers were synthesized for the purpose of the analysis, by dynamic and static light scattering, of the conformation and the compositional heterogeneity [70]. It was observed that the RAFT copolymers of molecular weight between 40 and 70 °K adopt cylindrical, rigid-rod all-trans conformation.

All the presented data regarding the synthesis of statistical copolymers of NVP with other comonomers via RAFT copolymerization are displayed in Table 1.

### 4.2. Block Copolymers Based on NVP Synthesized via RAFT Polymerization

#### 4.2.1. Introduction

The most important class of polymeric materials, which plays a central role in contemporary Polymer Science, is the family of block copolymers [71,72,73]. This family consists of chemically different polymeric chains (blocks) connected in linear arrangements. The incompatibility, which is frequently developed among the different blocks, gives rise to a rich variety of well-defined self-assembled nanostructured materials both in bulk [74,75,76] and in selective solvents [77,78]. These self-assembled structures are frequently responsive to external stimuli, such as temperature, pH, light irradiation and redox environment. Therefore, numerous applications ranging from thermoplastic elastomers to information storage, drug delivery and photonic materials [79,80,81,82,83,84,85,86,87] have appeared over the years.

A huge variety of polymeric techniques and methodologies have been employed for the synthesis of block copolymers [72]. An indispensable requirement for the preparation of tailor-made, well-defined block copolymer structures is the utilization of a living, or at least a controlled chain-growth polymerization technique, in association with suitable purification methods for all reagents employed (monomers, solvents, linking agents, additives, etc.) and methods to avoid the introduction of any impurity in the polymerization system. Taking these precautions, undesired termination or transfer reactions are minimized, thus leading to the synthesis of structures characterized by chemical and molecular homogeneity.

Recent progress in RAFT polymerization has allowed the synthesis of a variety of block copolymers [27,28,88,89,90]. The synthetic procedure has several specific features, due to the unique characteristics of the polymerization methodology. The control over the polymerization process in RAFT is obtained through an equilibrium between active and dormant chains, which is achieved via degenerative transfer. For this reason, an initial source of radicals, typically a conventional radical initiator, is required. RAFT is not a true living polymerization method, since it is accompanied by termination reactions. The extent of termination events can be predicted since it is connected with the number of radicals that are present in the system. By minimizing the amount of initiator, termination reactions are minimized as well but not completely prevented. Polymer chains with thiocarbonyl end-groups are considered as living since they can further promote polymerization reactions. On the other hand, chains without these end-groups are dead chains, as chain extension reactions are not possible. Moreover, two types of polymer chains are present in solution depending on the type of initiation, which promoted the polymerization, either by the radical initiator directly or the RAFT agent via the fragmentation process. The exact type of the α- or the ω-chain end will define, to a great extent, the ability of the polymer chain to promote the initiation of another monomer leading to the synthesis of block copolymers.

In the case of typical nitroxide-mediated (NMP) [91] and atom transfer radical polymerization (ATRP) [92,93], where reversible radical deactivation takes place, it is imperative to stop the polymerization at relatively low conversion if chain extension is desired (leading to increased molecular weights or the synthesis of block copolymers). High conversions limit the livingness of the system affording dead macromolecular chains. However, in RAFT polymerization, this is not a matter of concern, since the number of dead chains depends on the initial number of radicals produced by the initiator. Therefore, small quantities of initiator will afford a minor contamination with homopolymer chains.

#### 4.2.2. Sequential Addition of Monomers

The simplest way to produce block copolymers is by sequential addition of monomers. In the case of RAFT polymerization, the first monomer is polymerized under suitable experimental conditions leading to the formation of macromolecular CTA, usually called macro-CTA or macro-RAFT agent, which is able to polymerize the second monomer. An essential requirement for the success of this process is that the Z-group of the initial CTA should be able to control the polymerization of both monomers. If the RAFT agent has a different reactivity with the two monomers, then the level of control will be different for these constituents leading to blocks with different molar mass dispersities, uncontrolled molecular weights and increased chemical heterogeneity. The different efficiency of the chain transfer reactions will definitely increase the possibility of termination reactions, leading to the formation of homopolymers along with the desired block copolymer. Extended and time-consuming purification procedures will be necessary in this case to obtain pure products. NVP belongs to the family of LAMs, and therefore copolymerization with other LAMs seems to be feasible and able to produce well-defined block copolymers, since both monomers require the same CTA for controlled polymerization [94,95]. The situation is more challenging when NVP is combined with monomers belonging to MAMs, where different RAFT agents are appropriate for each monomer. In these cases, alternative approaches can be employed to promote the synthesis of the desired block copolymers.

Several experimental parameters may influence the efficiency of the synthetic approach towards the formation of well-defined block copolymers. Among them the most important are the following: (a) the order of monomer addition, (b) the effect of initiator concentration, (c) the solvent of the polymerization reaction, (d) the polymerization temperature, (e) the handling of the macro-RAFT agent and (f) the molecular weight of the macro-RAFT agent.

The order of monomer addition is essential in block copolymer synthesis, since the first block will serve as the R-group of the macromolecular RAFT agent. In order to promote the polymerization of the second monomer, this macro-R group should be a good homolytic leaving group in examination with the propagating radical of the second monomer and must efficiently initiate the polymerization of this monomer.

The employment of low initiator concentration reduces the possibilities of termination effects and defects during the chain extension reaction on the one hand, but on the other hand reduces the rate of polymerization. Therefore, a balance between these phenomena should be kept by carefully choosing the experimental reaction conditions. To avoid termination incidents and the formation of a high number of dead chains, it is imperative to keep the monomer conversion for the first block relatively low, e.g., up to 70%.

The choice of solvent is also important for the efficient synthesis of block copolymers, especially in the case where the two monomers lead to polymers with different solubilities, as in the case of amphiphilic block copolymers. PNVP is water soluble and also soluble in polar solvents. It is advised to employ common good solvents and if possible, the same solvent for the polymerization of both blocks.

The polymerization temperature should be chosen with care, since it affects the decomposition efficiency of the initiator and the rate of the polymerization, and it may result in the loss of the RAFT agent end-groups. Some of these groups are sensitive to light irradiation and pH and consequently, the handling of the macro-CTA agent should be employed with extra care.

Finally, the molecular weight of the macromolecular RAFT agent greatly influences the efficiency of the formation of block copolymers. Upon increasing the molecular weight of the first block, a substantial decrease in the polymerization rate of the second monomer was observed. In addition, incomplete chain extension was obtained leading to block copolymers with poor control over their molecular characteristics, increased chemical heterogeneity and high levels of contamination with homopolymers.

##### Block Copolymers with Low Activated Monomers (LAMs)

Several efforts have been described for the synthesis of block copolymers of PNVP with other LAMs. The most frequently reported case is the synthesis of amphiphilic block copolymers with poly(vinyl acetate), PVAc. Subsequent hydrolysis of PVAc may lead to the preparation of double hydrophilic copolymers of PNVP and poly(vinyl alcohol), PVA.

RAFT copolymerization was employed for the synthesis of PNVP-b-PVAc block copolymers [96]. NVP was polymerized first in 1,4-dioxane at 70 °C employing AIBN as initiator and S-2-propionic acid-O-ethyl xanthate as the CTA. The polymerization was terminated at a conversion of 80%, and the prepared polymer was precipitated in diethyl ether. The PNVP block was then employed as the macromolecular CTA for the polymerization of VAc in the next step, using AIBN as initiator. The polymerization was also conducted in 1,4-dioxane at 70 °C. The copolymers were analyzed by NMR spectroscopy, SEC, multiangle light scattering and MALDI-TOF mass spectrometry. Samples of rather broad molecular weight distribution were obtained. The dispersity was found to increase upon increasing the PVAc content. The copolymer composition by NMR measurements showed strong deviations from the theoretical predictions (molar ratios of monomers in feed). SEC analysis revealed the coexistence of both PNVP and PVAc homopolymers in the final products. These results were mainly attributed to side reactions during RAFT polymerization, leading to copolymers with various end-groups and indicating that the control of the copolymerization reaction was not at the highest level.

More detailed analysis employing additional chromatographic separation techniques was performed [97]. The experimental results during the polymerization of NVP revealed extensive chain transfer reactions to the polymerization solvent (Figure 4, Figure 5, Figure 6 and Figure 7). The solvent radicals were efficient in polymerizing the added monomers, leading to the formation of non-reactive homopolymer, as byproducts during the copolymerization process. In addition, dimers of NVP, esters of NVP, hydration of the double bond of NVP, thermal decomposition or hydrolysis of the xanthate end group were reported to take place during the synthesis. Similar problems were also obtained for the polymerization of VAc, as shown in Figure 4, Figure 5, Figure 6 and Figure 7.

A similar approach was adopted for the synthesis of amphiphilic PNVP-b-PVAc block copolymers. NVP was polymerized first at 60 °C, in bulk to avoid side reactions with the solvent [98]. AIBN was employed as initiator, whereas S-(2-cyano-2-propyl) O-ethyl xanthate as the CTA. The polymerization was allowed to take place for 6 h to minimize side reactions. The PNVP homopolymer was precipitated, dried and was then used as macromolecular RAFT agent for the polymerization of VAc. The polymerization of the second block took place in methanol as the solvent at 60 °C as well. The synthetic approach is depicted in Figure 8. Moderate dispersities (approximately 1.6–1.7) were measured by SEC. The micellization behavior in aqueous solutions of these copolymers was examined, and the supramolecular structures were employed as vehicles for drug delivery purposes.

A different methodology was employed for the synthesis of the PNVP-b-PVAc amphiphilic blocks [99]. As in the previous cases, NVP was polymerized first. The polymerization was conducted under various experimental conditions: (a) in dioxane using AIBN as initiator and S-4-(hydroxymethyl)-benzyl carbonodithioate as the CTA at 80 °C, (b) under similar experimental conditions but at 60 °C, (c) in bulk at 80 °C using the same initiator and CTA and (d) in aqueous solution at 25 °C using the same CTA but a redox initiating system consisting of t-butyl peroxide and ascorbic acid. It has been shown [100] that the end-dithiocarbonate group can be thermally eliminated during the polymerization reaction. However, under all these conditions, the xanthate functionality was kept rather low, below 55%. Only with bulk polymerization at short reaction times was a high functionality (equal to 86%) achieved. The data were obtained by NMR analysis on rather low-molecular-weight samples in order to facilitate the end-group analysis and receive the highest possible accuracy. The molecular mass dispersities were relatively low (around 1.3), indicating that RAFT is efficient for the polymerization of NVP. Nevertheless, side reactions cannot be excluded even in the case of redox initiation, leading to partial elimination of the desired xanthate end-groups. Subsequent emulsion polymerization of VAc was employed in aqueous solutions at ambient conditions, using the PNVP chains as the macromolecular RAFT agent and redox initiation with t-butyl peroxide and ascorbic acid. The block copolymerization was conducted over 48 h, as shown in Figure 9. Bimolecular distributions were obtained by SEC as a result of the low percentage of xanthate functionality of the PNVP chains. The pure block copolymers were obtained by centrifugation. Samples with much higher VAc contents were obtained in all cases. Consequently, the desired structures can be obtained by this approach but with very limited control over the molecular characteristics. Obviously, when the synthesis of block copolymers with higher NVP content is attempted, the purification of the final product will be much more difficult.

The synthesis of PNVP-b-PVAc block copolymers as precursors for the synthesis of double hydrophilic PNVP-b-PVA copolymers has been reported [101]. The synthesis was attempted starting either from the polymerization of NVP or of VAc. In both cases, 4,4′-azobis(4-cyanovaleric acid), ACVA, was employed as initiator, and a benzyl and an O-ethoxy functional xanthate as CTA (methyl (ethoxycarbonothioyl) sulfanyl benzene). The polymerization of NVP was conducted in dioxane at 70 °C for 3 h. The PNVP macro-CTA was further extended with the polymerization of VAc, using ACVA as initiator in 1,4-dioxane solutions at 68 °C for 12 h. Using the reverse procedure, VAc was initially polymerized in bulk at 68 °C for 10–30 min, depending on the desired conversion. The PVAc macromolecular CTA was further extended by polymerization of NVP employing ACVA as initiator, in 1,4-dioxane solutions at 70 °C for 12 h. Rather low monomer conversions were obtained in almost all cases in an effort to reduce the elimination of the xanthate end-group and increase the efficiency in the synthesis of the desired block copolymers. Indeed, SEC analysis revealed monomodal distributions with moderate molar mass dispersities. The samples had very low molecular weights and were subjected to quantitative hydrolysis employing a hydrazine hydrate aqueous solution (80% in water) for 5 h at 60 °C (Figure 10). The resulting PNVP-b-PVA block copolymers were found to serve as efficient antifreeze agents.

Several other examples for the synthesis of block copolymers of PNVP with LAMs other than PVAc have been reported in the literature. The synthesis of amphiphilic block copolymers of PNVP with poly(N-vinyl carbazole), PNVP-b-PNVK, was realized by RAFT polymerization and sequential addition of monomers [102]. Benzyl piperidine dithiocarbamate was synthesized (Figure 11) and employed as CTA for the preparation of the desired block copolymers. NVP was polymerized first at 60 °C in toluene solutions using AIBN as initiator. The molecular weight increased linearly with conversion, and the dispersity was rather constant, ranging within the values 1.3 to 1.4, indicating a relatively good control of the polymerization reaction. Conversions as high as 80% were obtained after 16 h of polymerization. The produced PNVP chains were subsequently employed as macro-CTA agents for the promotion of the polymerization of NVK leading to the synthesis of the corresponding block copolymers, as shown in Figure 12. The polymerization of NVK was carried out again in toluene solutions at 60 °C using AIBN as initiator. Although the copolymer dispersity was nearly the same as that of the PNVP block, the SEC trace was not symmetrical, indicating the presence of several byproducts. No attempt was made to clarify the issue of purity of the products. In all cases, the copolymers were rich in PNVP, probably to ensure their solubility in aqueous solutions. The micellization behavior of these amphiphilic copolymers was studied. It was found that these micelles have very low cytotoxicity and thus can be used in various biomedical applications, such as drug delivery.

Double hydrophilic block copolymers of PNVP with poly(N-vinyl caprolactam), PNVP-b-PNVCL, were synthesized using xanthates as CTA and AIBN as initiator [103]. The copolymerization was conducted at 60 °C in dioxane solutions, starting either from NVP or NVCL, as shown in Figure 13. In order to avoid transfer and termination reactions and achieve the preparation of macromolecular CTAs with the desired end groups, the monomer conversion was kept lower than 80%. SEC analysis revealed rather broad molecular weight distributions for the copolymers and, in certain cases, non-symmetrical peaks or even bimodal distributions, indicating that it is not always easy to avoid termination reactions. The produced block copolymers were temperature responsive, since PVCL shows a lower critical solution temperature, LCST, in the range of 31–38 °C.

The solution and mini-emulsion RAFT polymerization of vinyl chloride, VC, was carried out using 3,3,4,4,5,5,6,6,7,7,8,8,8-tridecafluorooctyl-2-(ethoxycarbonothionyl)thio propanoate as CTA and azobis(2,4-dimethylvaleronitrile) as the initiator [104]. The solution polymerization was conducted in 1,4-dioxane at 45 °C, whereas the miniemulsion polymerization was at 70 °C. Better control in the molecular characteristics was achieved in solution polymerization and by keeping the conversion lower than 50%. Chain extension with the polymerization of NVP afforded a block copolymer. Only one sample of this type was prepared with a very low-molecular-weight block of PNVP. It remains a challenge whether high-molecular-weight block copolymers can be prepared with this approach.

Stimuli-responsive block copolymers of PNVP and poly(3-ethyl-N-vinyl pyrrolidone), PNVP-b-PENVP, were synthesized by RAFT techniques (Figure 14) [105]. PENVP shows a sharp LCST at 26–27 °C, and thus the block copolymer is thermoresponsive. S-(2-cyano-2-propyl) O-ethyl xanthate was employed as the CTA, AIBN as the initiator and the polymerizations were conducted in 1,4-dioxane solutions at 60 °C. Monomodal and symmetrical peaks of rather low dispersity were obtained from the polymerization of both monomers. In addition, the molecular weights were in rather close proximity to the theoretical values. However, it was found that the xanthate chain end-groups were labile. Up to 30% of these groups were eliminated, leading to the formation of unsaturated chain ends (Figure 15). Therefore, the subsequent synthesis of block copolymers, starting from either the PENVP or the PNVP chains, leads to several byproducts, as can be seen from the tailing effects and the bimodal distributions in SEC traces. The resulting block copolymers were thermoresponsive in aqueous solutions, leading to the formation of various supramolecular structures, such as shperical and cylindrical micelles or vesicles. The solution concentration and the copolymer composition also play an important role in the self-assembly process. The biocompatibility of both components of the block copolymers allows the system to be applied as a drug delivery vehicle.

Vinyl ethers are typical electron-donating monomers and therefore are susceptible to cationic polymerization. However, when the side group of the monomer is an electron-withdrawing group, conventional or controlled radical polymerization can be carried out. Along these lines, the RAFT polymerization of 2-hydroxyethyl vinyl ether, HEVE, was successfully attempted using methyl(phenyl)carbamodithioate as the CTA [106]. The polymerization was conducted in bulk at 70 °C using dimethyl 2,2′-azobis(isobutyrate), V-601, as radical initiator. It is imperative to use CTAs without protonic acid moieties in order to avoid the formation of acetals. PHEVE homopolymers of relatively low molecular weights and moderate values of dispersity (Ð < 1.4) were prepared at conversions lower than 50%. This is an indication that at low conversions there is a reasonable degree of control, but upon increasing the conversion, chain transfer or termination reactions take place, avoiding the further increase in conversion. Using these PHEVE homopolymers as macromolecular CTAs for the polymerization of NVP, a block copolymer PHEVE-b-PNVP was obtained. The conversion of polymerization of NVP was as high as 90%. However, the dispersity of the block copolymer was very large (Ð = 2.31), indicating the presence of several side reactions during the copolymerization process (Figure 16). Nevertheless, this is a very interesting work, opening new horizons with the controlled radical polymerization of functional vinyl ethers and their combination with other radically polymerized monomers.

Several trithiocarbonates were employed as CTAs to promote the RAFT polymerization of 1,2,4-triazolium salts, such as N-vinyl-4-ethyl-1,2,4-triazolium bromide, NVETri-Br (Figure 17) [107]. When the polymerization was conducted in methanol at 60 °C for 24 h using AIBN as initiator, quantitative conversions and controlled molecular characteristics were obtained. Taking advantage of these results, the synthesis of the PNVP-b-PNVETri-Br block copolymer was attempted, as shown in Figure 18. NVP was polymerized first in bulk, employing O-ethyl-S-(1-ethoxy carbonyl) ethyl dithiocabonate (CTA3) and AIBN at 60 °C for 50 min. A low-molecular-weight product with very narrow molecular weight distribution in quantitative yield was obtained. Subsequent addition of NVETri-Br to the originally prepared PNVP macromolecular CTA afforded the desired block copolymer. The copolymerization took place in methanol at 60 °C over a period of 24 h. The final products were purified either by precipitation in acetone/chloroform (7/3 vol%) or by dialysis with methanol and then by reprecipitation in acetone/diethyl ether (6/4 vol.%). The purification was necessary, since PNVP homopolymers were traced in the final products. After this procedure, monomodal and symmetrical peaks of relatively broad molecular weight distribution were obtained for the block copolymers. The reverse mode of monomer addition failed to give the desired structures. These copolymers were subjected to ion exchange reactions, leading to products with high conductivity.

Poly(vinylidene fluoride)-b-PNVP, PVDF-b-PNVP, block copolymers were synthesized by RAFT polymerization [108]. The polymerization of VDF was conducted first in dimethyl carbonate, at 73 °C for 20 h using O-ethyl-S-(1-methoxycarbonyl)ethyldithiocarbonate as the CTA and tert-amyl peroxy-2-ethylhexanoate as the initiator. The reaction took place in an autoclave. The conversion was very low (less than 20%). However, the dispersity was relatively narrow. The xanthate-terminated PVDF served as the macromolecular CTA for the polymerization of NVP. This second polymerization step was performed in N,N-dimethylacetamide at 70 °C for 24 h using AIBN as the radical initiator (Figure 19). The reaction conversion was almost quantitative, and the final copolymers had relatively narrow molecular weight distributions. Only low-molecular-weight samples were prepared in this study. Therefore, definite conclusions on the properties cannot be given. However, it was shown that microphase separation in bulk was not observed due to the compatibility of the constituent components of the block copolymer, whereas self-assembly takes place in aqueous solutions, leading to micellar formation.

##### Block Copolymers with More-Activated Monomers (MAMs)

The synthesis of block copolymers of NVP with MAMs is very challenging, as already explained. However, progress in RAFT polymerization has allowed the synthesis of rather well-defined block copolymers consisting of monomers with different reactivities. This can be achieved by two different synthetic strategies [27,88,90]. These involve the use of novel CTAs (called universal CTAs), which are able to control the polymerization of both LAMs and MAMs, and the use of CTAs that can efficiently polymerize both types of monomers after simple chemical transformations, e.g., after protonation in acidic environment. These are called switchable CTAs. Both methodologies have been employed for the synthesis of block copolymers of NVP with MAMs.

It has been proven that several xanthates and dithiocarbamates may offer good control over the polymerization of LAMs and at the same time relative control over the polymerization of MAMs [95]. Xanthates can be efficiently employed for the synthesis of block copolymers through sequential addition of monomers combining LAMs with MAMs and showing higher activity, such as acrylates and acrylamides. Rhodixan A1 is a typical example of such a CTA (Figure 20). Dithiocarbamates are efficient in controlling the polymerization of styrene, acrylates and acrylamides, while at the same time they are versatile for the polymerization of LAMs, thus leading to the synthesis of rather well-defined block copolymers. The key factor of this CTA as a universal agent is to conjugate the lone pair of electrons of the nitrogen atom with carbonyl or aromatic groups. Characteristic examples are also given in Figure 20.

Switchable CTAs are reagents with reactivity, which can be modulated upon changing the external conditions. Characteristic examples are given in Figure 21 with the pH-sensitive N-aryl-N-(4-pyridinyl)dithiocarbamate. The reactivity of the thiocarbonyl group is switched upon protonation of the pyridinium ring, as shown in Figure 22. In the neutral form of the CTA, the C=S bond has low reactivity, thus promoting the polymerization of LAMS, whereas protonation of pyridine, either by a strong acid, such as trifluoromethanesulfonic acid or p-toluenesulfonic acid, or a non protic Lewis acid, such as aluminum triflate, activates the C=S bond, thus promoting the polymerization of MAMs. Under suitable experimental conditions, well-defined poly(MAM)-b-poly(LAM) block copolymers can be achieved (Figure 23). Two important prerequisites for the successful synthesis of the desired products are the following: (a) the MAM should be polymerized first due to the relative homolytic leaving group ability of MAMs and LAMs derived propagating radicals and (b) extra care should be given to avoid the presence of any remaining quantity of the acid in the first block, when switching to polymerize the LAM. Otherwise, the control over the polymerization of LAM is very limited.

Alternative ways to switch the reactivity of the CTAs have been reported in the literature [109,110,111,112].

##### Employment of Universal CTAs

Amphiphilic or potentially double hydrophilic block copolymers of PNVP and poly(2-vinyl pyridine), PNVP-b-P2VP, were prepared via RAFT. NVP was initially polymerized in THF solution at 80 °C employing AIBN as the radical source and S-1-dodecyl-S’’-(α,α’-dimethyl-α’’-acetic acid)trithiocarbonate as the CTA [113]. Subsequent addition of 2VP and a new amount of AIBN after heating at 75 °C in DMF solution resulted in the formation of the desired block copolymer (Figure 24). The conversion of the NVP polymerization was kept low (~50%) to ensure the quantitative presence of the end-groups and controlled polymerization. SEC analysis revealed that the first block reacted quantitatively with the second monomer to afford the block copolymer. Tailing effects or shoulders were absent from the SEC traces. The dispersity values were relatively low and stable (Ð = 1.50) for both the first block and the final copolymer. This approach was therefore efficient for the preparation of these block copolymers.

The synthesis of block copolymers with PNVP as the first block and PDMAEMA or poly(styrene-alt-maleic anhydride) as the second block were reported, employing S-benzyl dithiobenzoate, BTBA, as the common CTA for both blocks and AIBN as the initiator [114]. The polymerization reactions took place at 80 °C in 1,4-dioxane solutions. NVP was always polymerized first, as shown in Figure 25. The copolymers were characterized by FTIR and NMR spectroscopy and by SEC. However, the SEC traces were not provided, and thus conclusions regarding the purity of the final products could not be made. In addition, no comments were made regarding the chemical purity of the copolymers. These oppositely charged block copolymers may self-assemble in aqueous solutions, forming stable spherical polyion complex micelles, which are responsive in the solution pH. The release profiles of coenzyme A were studied in different pH values.

Block copolymers of PNVP with poly(N,N-diethyl acrylamide), PDEAM were synthesized by the RAFT methodology and sequential addition of monomers [115]. DEAM was polymerized first using trithiocododecanoic acid-2-cyanoisopropyl as the CTA (Figure 26). The reaction took place in ethyl acetate at 70 °C, employing AIBN as the initiator. The produced PDEAM served as the macromolecular CTA for the subsequent polymerization of NVP. The reaction was conducted in acetone solution. In all cases very low-molecular-weight samples were obtained (M_n_ < 4 × 10^3^), whereas the dispersity values were reasonably low. However, detailed SEC analysis from the first block and the final products was not reported, and therefore, comments about the chemical homogeneity of the copolymers cannot be stated. The thermo-sensitivity of the block copolymers was studied by measuring the lower critical solution temperature in aqueous solutions.

Block copolymers PNVP-b-PDMAEMA were also prepared following a different protocol. NVP was polymerized first in DMF solutions at 60 °C using AIBN as initiator and S-(2-ethyl proprionate)-O-ethyl xanthate as the CTA [116]. The PNVP-CTA, thus formed, was subsequently employed for the block copolymerization of DMAEMA under the same experimental conditions to afford the desired block copolymers (Figure 27). Only one sample of rather low molecular weight was prepared, with a surprisingly narrow molecular weight distribution. However, SEC traces monitoring the synthesis were not provided to further support the reported conclusions. The formation of polyplexes with DNA was examined, showing rather high transfection efficiency.

PMMA-b-PNVP block copolymers were synthesized by RAFT and sequential addition of monomers starting from the polymerization of MMA [117]. The polymerization took place in toluene solution at 68 °C using AIBN as the initiator and isopropylxanthic disulfide as the CTA. The conversion of the reaction was kept rather low (less than 35%) in order to achieve the quantitative presence of the end-groups of the PMMA chains, so that it would be efficiently employed as the macromolecular CTA in the next step. Subsequent addition of NVP afforded the desired block copolymer. The polymerization was allowed to take place at 75 °C up to full conversion (Figure 28). The reaction sequence was monitored by SEC, indicating that the procedure is free of termination reactions and the products had relatively low dispersity values. The reverse procedure, i.e., the polymerization of NVP first followed by the polymerization of MMA later, was not proven efficient. The block copolymer was subsequently incorporated into polybenzoxazine, leading to the formation of nanostructured thermosets via a reaction-induced separation mechanism.

The synthesis of double hydrophilic block copolymers PNVP-b-poly(2-acrylamido-2-methyl-1-propanesulfonic acid), PNVP-b-PAMPS, has been described [118]. In this case, NVP was initially polymerized followed by polymerization of the second monomer. The polymerization reaction of NVP was performed in 1,4-dioxane at 70 °C using AIBN as the initiator and 2-dodecylsulfanylthiocarbonylsulfanyl-2-methyl propionic acid, DMP, as the CTA. The PNVP chains were then employed as the macromolecular CTA for the polymerization of AMPS in DMF solutions in the presence of AIBN. The synthesis is given in Figure 29. The one sample that was synthesized was characterized by NMR and IR spectroscopies and SEC. Mixing of this block copolymer with PNVP-b-PDMAEMA block leads to the spontaneous formation of pH-sensitive polyion complexes. The encapsulation of folic acid and the release profiles of this substance were studied under different pH values.

Another type of hydrophilic block copolymer consisting of PNVP and poly(methacrylic acid), PMAA, blocks was prepared by RAFT [119]. As in the previous case, NVP was polymerized first in ethanol solution at 65 °C with AIBN as initiator and 1-phenylethyl dithiobenzoate as the CTA. The sodium salt of MAA was then added to the PNVP macromolecular CTA in ethanol solution. The reaction took place at 85 °C in the presence of AIBN. The conversion of the MAA sodium salt was less than 50%. The dispersity of the final block copolymer (Ð = 1.42) was substantially increased compared to the dispersity of the first block (Ð = 1.26), indicating that the MAA polymerization cannot be very well controlled (Figure 30). These double hydrophilic copolymers were employed as effective crystal growth modifiers of CaCO_3_ particles, leading to various morphologies upon varying the copolymer concentration.

A novel redox initiation system based on sodium sulfite and tert-butyl hydroperoxide in combination with O-ethyl-S-(1-methoxycarbonyl) ethyl dithiocarbonate (Rhodixan A_1_) as the CTA were employed for the synthesis of linear PNVP [120]. Following this procedure, double hydrophilic block copolymers consisting of PNVP as the second block and poly(acrylamide), PAm, poly(acrylic acid), PAA, poly(sodium 2-acrylamido-2-methylpropanelfunate), PAMPS, along with poly(3-acrylamidopropyltrimethyl ammonium chloride), PAPTAC, as the first block were synthesized (Figure 31). The macromolecular CTA was obtained in a mixture of water and ethanol using 4,4′-azobis(4-cyanovaleric acid), ACVA, as the initiator. The polymerization was conducted at 60 °C under argon atmosphere for 3 h. The conversion was almost quantitative, and the molecular weight of the macro-CTA was in all cases very low in order to facilitate the solubility of the final products. Subsequent addition of NVP was performed to yield the desired block copolymers. The polymerization of NVP took place in aqueous solution using sodium sulfite and tert-butyl hydroperoxide as the initiating system at room temperature. SEC analysis was not very efficient due to the strong adsorption of the copolymers on the separation columns. Therefore, DOSY NMR experiments were conducted to undoubtedly verify the successful synthesis of the block copolymers.

Amphiphilic block copolymers with PNVP and polyisoprene, PI, blocks were synthesized by sequential addition of monomers and RAFT polymerization methodologies [121]. NVP was initially polymerized at 80 °C in the presence of ACVA as the initiator and S-1-dodecyl-S’-(α,α’-dimethyl- α’’-acetic acid)trithiocarbonate as the CTA. Products of relatively broad dispersity at rather low conversions (about 50%) were obtained. The block copolymerization was conducted upon addition of isoprene to the PNVP macro-CTA in the presence of tert-butyl peroxide as initiator at 125 °C for 24 h (Figure 32). The polymerization yields were very low and the dispersity values extremely high. SEC analysis revealed the presence of bimodal distributions in almost all cases showing that the reaction is not well controlled, and the final products are ill-defined. These copolymers were further subjected to cross-linking procedures, after reaction with sulfur monochloride to obtain complex amphiphilic networks.

Benzyl ethyl trithiocarbonate as CTA and AIBN as initiator were employed for the RAFT polymerization of n-butyl acrylate, BuA [122]. The reaction was performed in 1,4-dioxane at 85 °C. Subsequent addition of NVP to the macromolecular PBuA CTA under the same experimental conditions afforded the desired amphiphilic block copolymers, PnBuA-b-PNVP, as shown in Figure 33. Although the molecular weight distributions were relatively narrow for the first block and the final copolymers, SEC analysis revealed tailing effects during the polymerization. This result was attributed to the gradual loss of the CTA’s trithiocarbonate moieties, as a result of the presence of chain transfer reactions to the monomer. It was found that the toxicity of these block copolymers is very low, thus allowing their employment in biomedical applications.

2-Cyanoprop-2-yl-1-dithionaphthalene, CPDN, is a well-known CTA providing very good control over the RAFT polymerization of MAMs. However, for LAMs, it fails to afford the same control. Nevertheless, it was found that employing 1,1,1,3,3,3-hexafluoro-2-propanol (HFIP) as the solvent, the polymerization of LAMs, such as NVP, can be promoted leading to well-defined products [123]. The reason for this change is the formation of hydrogen bonds between NVP and HFIP. This was confirmed both by NMR analysis and simulation techniques. A redistribution of the double bond electrons is accomplished so that the LAMs could behave as more active monomers. Under these experimental conditions, NVP behaves as an MAM-like monomer and therefore, its activity matches that of the CTA, which in turn becomes a universal CTA, efficient for polymerization of both types of monomers. Therefore, the efficient synthesis of PNVP-b-PS and PNVP-b-PMAA (PMAA is poly(methyl acrylate)) can be achieved. The final products have a reasonably narrow molecular weight distribution. However, SEC analysis revealed that termination and chain transfer reactions cannot be completely avoided, leading to up to 25% dead chains during the copolymerization reaction.

A series of four CTAs was employed for the synthesis of alternating copolymers between VAc and tert-butyl-2-trifluoromethacrylate (MAF-TBE) [124]. In particular, the following CTAs were tested: cyanomethyl-3,5-dimethyl-1H-pyrazole-1-carbothioate (CDPCD), O-ethyl-S-(1-methoxycarbonyl) ethyl dithiocarbonate (CTA-XA), 2-cyano-2-propyl benzothioate (CPDB) and 4-cyano-4-(2-phenylethanesulfanylthiocarbonyl)sulfanyl pentanoic acid (PETTC), as shown in Figure 34. The copolymerization was conducted in bulk at 40 °C with 2,2′-azobis-(4-methoxy-2,4-dimethylvaleronitrile) (V-70) as the radical source. CDPCD was proven to be the most effective CTA in controlling the molecular characteristics of the copolymers. CPDB and PETTC failed to produce copolymers due to the fact that the Z group of these CTAs stabilize the VAc-based intermediate radical so much, finally causing complete inhibition. ^1^H- and ^19^F-NMR spectroscopies were employed to confirm the alternating structure of the copolymer. It is interesting to note that VAc is a typical LAM, whereas MAF-TBE may be considered as an MAM, due to the radical stabilizing effect of the carbonyl group. The P(VAc-alt-MAF-TBE) copolymers were then employed as macro-CTAs for the block copolymerization NVP, leading to amphiphilic block terpolymers (Figure 35). However, the control was not very efficient, since products of high dispersity were obtained (Ð = 1.9). Actually, the SEC trace showed a bimodal distribution. This result may be attributed to the experimental conditions (40 °C, DMF as the solvent) under which the polymerization was conducted. The reverse polymerization, i.e., first the polymerization of NVP and then the copolymerization of VAc and MAF-TBE, was not successful either.

O-phenyl-S-[1-(phenylethyl)] dithiocarbonate was synthesized (Figure 36) and, in an elegant way, was employed both as a CTA and initiator for the RAFT polymerization of NVP and several MAMs [125]. Under UV or visible light irradiation, a series of MAMs (styrene, butyl acrylate and methyl acrylate) was efficiently polymerized in bulk at room temperature. Subsequent addition of NVP to these macro-RAFT agents afforded PMAM-b-PNVP block copolymers in the absence of any photocatalyst and initiator. The copolymerization was again conducted in bulk at room temperature. The polymerization conversion of MAMs was very high, and very good control over the molecular characteristics was achieved. The copolymerization with NVP revealed the presence of monomodal SEC traces. However, the dispersity values were much higher than those measured for the PMAM block, and in addition, the SEC peaks were not always symmetrical, indicating the formation of dead chains of the first block or during the initial stages of copolymerization due to the irradiation with UV or visible light.

##### Employment of Switchable CTAs

pH-switchable CTAs have been initially introduced in the literature, as shown in Figure 37 [126,127]. The neutral form of N-methyl-N-(4-pyridinyl) dithiocarbamates provides excellent control over the polymerization of LAMs, including NVP. The protonated form of the same compound, which is formed by addition of one equivalent of strong acid (such as 4-toluenesulfonic acid or trifluoromethanesulfonic acid) is effective in the polymerization of MAMs. When less than one equivalent of the strong acid or a weaker acid (such as acetic acid) was employed, it led to poor control over the polymerization of MAMs. Nonprotic Lewis acids (e.g., aluminum triflate) were also very effective. The most efficient method for the synthesis of the desired block copolymers is to start with the polymerization of the MAM followed by the polymerization of LAM, thus leading to polyMAM-b-polyLAM block copolymers. An example for the synthesis ofpoly(MMA-b-VAc) is given in Figure 37.

Using this protocol, block copolymers of poly(N-dimethylacrylamide), PDMAm and PNVP were prepared [128]. The polymerization of DMAm was conducted for 30 min in aqueous solution at 80 °C, employing cyanomethyl methyl(pyridin-4-yl)carbamodithioate, as the CTA, 4-toluenesulfonic acid, TsOH, and 2,2′-azobis[2-methyl-N-(2-hydroxyethyl)propionamide], VA086, as the radical initiator. The solution was then neutralized with excess sodium bicarbonate and the final PDMAm-macro-CTA was obtained after removal of the water by lyophilization. Polymers of controlled molecular weights and narrow molecular weight distributions were produced. The polymerization of NVP was then performed in organic solvents, since in aqueous solutions, hydrolytic instability of the hemithioaminal group is observed. Care should be taken to avoid any excess of base or acid in the system. Under these conditions, well-defined block copolymers are synthesized (Figure 38).

A similar approach was followed for the synthesis of PtBuMA-b-PNVP block copolymers [129]. A pH-switchable CTA was employed in this case as well, similar to that reported for the synthesis of the PDMAm-b-PNVP blocks. In particular, 2-cyanopropan-2-yl N-methyl-N-(pyridine-4-yl)carbodithioate was employed as the CTA, along with its derivative bearing a N-hydroxysuccinimide terminal functional group (Figure 39). The protonated form of the CTA promotes the controlled polymerization of tBuMA in dioxane solution at 70 °C using AIBN as the radical initiator. The protonation of the CTA took place using trifluoromethane acid. Care should be given to the amount of acid employed. Using less than the stoichiometric amount results in poor control over the polymerization of the methacrylate, whereas using more than the stoichiometric amount leads to hydrolysis of the RAFT agent and the tert-butyl side groups. The protonated macro-RAFT agent was then neutralized by sodium carbonate and was employed to promote the polymerization of NVP in the presence of AIBN at 70 °C in dioxane solution, leading finally to the synthesis of PtBuMA-b-PNVP block copolymers. Subsequent acidic hydrolysis of the tert-butyl groups provides the double hydrophilic copolymer PMAA-b-PNVP. The micellization behavior of these block copolymers was studied in acidic aqueous solutions. It was shown that these systems can be potentially applied as drug and/or protein delivery vehicles.

Amphiphilic block copolymers with polystyrene, PS or poly(2,3,4,5,6-pentafluorostyrene), PPFS, as the hydrophobic block and PNVP as the hydrophilic block were synthesized by RAFT and the use of the switchable CTA 2-cyanopropan-2-yl N-methyl-N-(pyridine-4-yl)dithiocarbamate, as shown in Figure 40 [130]. The protonation of the CTA was achieved in the presence of trifluoromethanesulfonic acid. The styrenic monomer was polymerized first using the protonated form of the CTA in bulk at 70 °C using AIBN as initiator. At conversions higher than 45%, irreversible termination or chain transfer reactions were observed, leading to gradual loss of the control over the molecular characteristics. The macro-RAFT CTAs were then deprotonated and employed for the polymerization of NVP, leading to the final amphiphilic block copolymers. The conversion can be quantitative. However, at conversions higher than 55%, a broadening of the polydispersity was observed. Nevertheless, high-molecular-weight PNVP blocks up to 300–400 kg mol^−1^ could be prepared. These copolymers were employed as effective kinetic hydrate inhibitors to the pure methane-water system.

Several N-methyl-N-4-pyridinyldithiocarbamates have been synthesized and examined as switchable CTAs for the polymerization of MAMs and NVP [131]. The structures are provided in Figure 41. Kinetic analysis showed that reagent **2** provided better control over the polymerization of NVP when this is conducted in 1,4-dioxane at 60 °C. Taking these results into account, the synthesis of poly(N-isopropylacrylamide)-b-PNVP, PNIPAM-b-PNVP, block copolymers was attempted, following the reaction Figure 42. NIPAM was polymerized first in 1,4-dioxane at 60 °C solution using the protonated form of the CTA and AIBN. Deprotonation of the macro-RAFT CTA was performed by NaHCO_3_, followed by the polymerization of NVP. A narrow molecular weight distribution PNIPAM-b-PNVP sample was obtained. However, the monomer conversions were kept relatively low to avoid termination and transfer reactions.

#### 4.2.3. Combination of Different Polymerization Techniques

The combination of different polymerization techniques for the synthesis of block copolymers opens new horizons in Polymer Chemistry, since monomers that cannot be polymerized with the same methodology are chemically forced to co-exist in the same structure. Therefore, novel products with unique properties can be obtained. For the efficient application of this approach, it is important to find suitable monomers and pinpoint the monomer addition order and experimental conditions of the copolymerization reactions. It is not always easy to control all the parameters that may influence the molecular characteristics, the chemical homogeneity and the purity of the block copolymers. The most efficient ways leading to the synthesis of block copolymers of PNVP with other suitable monomers will be reported in the following sections.

##### Dual Functions CTAs (Inifers)

The employment of heterofunctional initiators, i.e., initiators bearing two or even more functional initiation sites, capable of the initiation of different polymerization mechanisms is well known in the literature [132,133]. Numerous block copolymers have been prepared by combining two mechanistically incompatible monomers. This can be achieved either by sequential polymerization of monomers or in a one-step procedure provided that the same experimental conditions can be applied for each reaction mechanism and the kinetics of polymerization can be well controlled for both monomers.

The combination of RAFT polymerization with other polymerization techniques in this sense can be accomplished employing functional CTAs, that is molecules that can act both as typical RAFT CTAs and as initiators. These compounds are usually called inifers, since they act both as initiators and as chain transfer agents.

Along these lines, a combination of RAFT with ATRP was employed for the synthesis of PNVP-b-PS, PNVP-b-PMMA and PNVP-b-PMA (MA is methyl acrylate) block copolymers [134]. For this purpose, dual function inifers were synthesized and applied for successive RAFT of NVP and ATRP reactions of the other monomers. Specifically, S-[1-methyl-4-(6-chloropropionate)ethyl acetate] O-ethyl dithiocarbonate (CPX) and S-[1-methyl-4-(6-chloroisobutyrate)ethyl acetate] O-ethyl dithiocarbonate (CiBX) were employed (Figure 43). The use of the corresponding bromoxanthate derivatives was excluded due to the extensive dimerization reaction of the NVP monomers, as a result of the S_N_2 reaction between the bromoester and the NVP.

Both CTAs promoted the well-controlled polymerization of NVP with conversions up to 70% and dispersities lower than 1.40. A characteristic feature of the polymerization was the pronounced induction period, which was equal to 8 and 3 h for CiBX and CPX, respectively. This behavior is rather common for LAMs and can be attributed to the slow fragmentation of the intermediate radicals during the polymerization reaction.

Subsequent chain extension was attempted with the ATRP of St and MMA. It was found that the xanthate end-group of the PNVP block was not able to cause chain transfer reactions with the growing styryl or methacrylate radicals, since these monomers belong to MAMs and xanthates are not efficient CTAs for these monomers. Therefore, the ATRP can be promoted without any interference of the xanthate end-group. Since the NVP moieties may act as chelating agents for metal ions, the strong complexing ligand tris[(2-pyridyl)methyl]amine, TPMA, was employed for the ATRP of St and MMA. The polymerization was conducted in relatively mild conditions (heating at 60 °C) to avoid side reactions, such as the elimination of the end-groups. Well-defined block copolymers were finally obtained with relatively narrow molecular weight distribution. However, the results were not very satisfactory for MA, since the xanthate end-group may be involved in the polymerization reaction, leading to multimodal SEC traces and ill-defined structures.

The alternative polymerization sequence, i.e., first the ATRP followed by the RAFT reaction was also tested. It was proven that this approach is less effective with rather low conversion during the ATRP reaction and with the preparation of ill-defined products with broad molecular weight distributions.

RAFT and Activators Re-Generated by Electron Transfer ATR (ARGET-ATRP) polymerization techniques were combined for the synthesis of block copolymers consisting of PNVP and poly(triethylene glycol methacrylate), PTEGMA [135]. For this purpose, S-[1,2-dimethyl-4-(6-chloroisobutyrate)ethyl acetate] O-ethyl dithiocarbonate was used as inifer, shown in Figure 44. NVP was initially polymerized in 1,4-dioxane at 65 °C using AIBN as the radical source. To this PNVP macroinitiator TEGMA, CuCl_2_ and the ligand TPMA were added in an isopropanol–water mixture (50% *v*/*v*). Ascorbic acid, ASCA, was finally added to initiate the polymerization at 40 °C (Figure 45). The polymerization of NVP was well controlled, leading to rather narrow molecular weight distributions. The SEC traces of the final block copolymers revealed the presence of a small amount of unreacted or terminated PNVP block and much broader molecular weight distributions. Large spherical micelles were obtained in aqueous solutions above the cloud point temperature. These self-assembled nanostructures were further stabilized even at lower temperatures after crosslinking the PNVP chains with potassium persulfate.

A CTA bearing a functional hydroxyl group, namely 2-hydroxyethyl 2-(ethoxycarbonothioylthio)propanoate, HECP, was applied as an inifer to promote the Ring-Opening Polymerization, ROP, of ε-CL and the RAFT statistical copolymerization of N-vinyl caprolactam, VCL, and NVP yielding finally the P(ε-CL)-b-P(VCL-stat-NVP) amphiphilic terpolymer, as shown in Figure 46 [136]. ROP of ε-CL was conducted first in anisole solution at 30 °C using diphenyl phosphate, DPP, as the catalyst, followed by the RAFT copolymerization of VCL and NVP employing 2,2′-azobis(4-methoxy-2,4-dimethyl valeronitrile), V-70, as initiator again at 30 °C. Very good control and extremely low dispersity values were obtained for the initial P(ε-CL) block and the final terpolymers. Thermoresponsive spherical micellar structures were obtained in aqueous solutions. It was found that the lower critical solution temperature could be tuned by choosing the specific architecture and the composition in the three components, thus providing the possibility to employ these terpolymers in various biomedical applications.

HECP was also used for the synthesis of P(ε-CL)-b-PNVP amphiphilic block copolymers (Figure 47) [137]. The synthesis was performed under similar experimental conditions, as previously mentioned, in a one-pot procedure. The conversion of ε-CL was equal to 50% when NVP was added to complete the copolymerization. Narrow molecular weight distribution products were obtained in all cases.

A similar combination of ROP and RAFT was employed for the synthesis of PNVP-b-PLLA block copolymers using 2-(N,N-diphenylcarbamothioylthio) propanoate (HDPCP) as the inifer [138]. The reaction sequence is displayed in Figure 48. The synthesis was performed in a one-step procedure. The mixture of NVP, LLA HDPCP, AIBN and DMAP (4-dimethylamino pyridine, which is the ROP catalyst) was dissolved in anisole and heated at 60 °C. The polymerization was monitored by SEC, revealing a very good control, and led to products with very narrow molecular weight distributions.

##### Transformation of the End-Group of the 1st Block to CTA, Suitable for the Polymerization of NVP

Another approach towards the synthesis of block copolymers involves the transformation of the end-group of polymer chains to CTAs, which are capable of promoting the RAFT polymerization. In this way, the transformed initial polymer plays the role of a macro-CTA and becomes the first block of the final copolymer structure. Compared to the sequential addition of monomers, this procedure has the advantage that the use of a polymer chain, which cannot be produced by RAFT polymerization as the first block, is possible. Therefore, a variety of structures can be obtained with very interesting and unique properties either in solution or in bulk. However, for the successful application of the methodology, there are two requirements: (a) the first block should be quantitatively functionalized with specific end-groups that will be subsequently transformed to CTAs. In the case of partial functionalization, this procedure will lead to a mixture of block copolymer and the first unfunctionalized block. The purification of this mixture is a time-consuming and frequently difficult process. (b) The transformation reaction should be quantitative and easy to perform under mild conditions in order to avoid possible damage (thermal decomposition, degradation, crosslinking etc.) of the initial block. This requirement limits the application of this approach to rather low-molecular-weight initial polymers. The higher the molecular weight of the first block, the less quantitative the transformation reaction will be.

This procedure has been efficiently applied to RAFT polymerization and in particular for the synthesis of block copolymers bearing PNVP as the second block. The common methodology involves the transformation of end-hydroxyl groups of various polymers to CTAs, which are suitable to promote the polymerization of NVP.

Following this method, PEG-b-PNVP block copolymers have been synthesized [139]. The reaction series is given in Figure 49. Commercially available low-molecular-weight, semitelechelic poly(ethylene glycol), PEG, sample with one functional end-hydroxyl group was transformed to two different macro-CTAs named as PEG-X1 and PEG-X2. PEG-X1 was prepared after reaction of PEG-OH with 2-bromopropionyl bromide followed by reaction with potassium O-ethyl xanthate. In a similar way, PEG-X2 was synthesized from the esterification reaction of PEG-OH with α-chlorophenylacetyl chloride and subsequent reaction with potassium O-ethyl xanthate as well. Using PEG-X1 and PEG-X2, the polymerization of NVP was attempted in THF solutions at 60 °C with AIBN as the radical source. It was observed that PEG-X2 failed to produce block copolymers. It seems that the phenyl group of PEG-X2 stabilizes the formed radical to such an extent that almost complete inhibition is achieved. On the other hand, PEG-X1 was proved a very efficient macro-CTA for the controlled polymerization of NVP and the synthesis of well-defined double hydrophilic block copolymers. 

A combination of ROP and RAFT following the same chemical transformation route was adopted for the synthesis of PCL-b-PNVP block copolymers, as shown in Figure 50 [140]. εCL was polymerized in a typical ROP reaction in toluene at 90 °C, employing n-butanol and Sn(Oct)_2_ as the initiating system. Different PεCL homopolymers with hydroxyl end-groups varying in molecular weights and with relatively low dispersity were obtained. Although the reactions were performed up to almost quantitative conversions transesterification side reactions were not mentioned. However, the SEC trace of one of the homopolymers revealed a pronounced shoulder, indicating that these side effects were not completely excluded. The end-OH groups were then transformed to CTA moieties after esterification with 2-bromopropionyl bromide followed by reaction with potassium O-ethyl xanthate, as previously mentioned. The polymerization of NVP was finally conducted in anisole solutions at 30 °C using 2,2′-azobis(4-methoxy-2,4-dimethyl valeronitrile) as initiator. SEC analysis confirmed that well-defined block copolymers of various molecular weights and compositions were obtained.

Similar PεCL-b-PNVP products were also prepared by another research group, following the reaction series of Figure 51 [141]. Compared to the previously reported procedure, the εCL was polymerized using benzyl alcohol instead of n-butanol and the polymerization temperature was higher (110 °C instead of 90 °C). On the other hand, NVP was polymerized in THF solutions at 80 °C using AIBN as radical initiator. It was found that for molecular weights up to 15,000, the control over the polymerization of the lactone was excellent. For higher molecular weights, broader or even bimodal distributions were obtained. The efficiency of the end-group transformation reactions for the synthesis of the macro-CTA was found to depend on the molecular weight of the polymer. The lower the molecular weight, the more efficient the functionalization reaction is. The polymerization of NVP was very well controlled up to 76% conversion. The presence of PεCL and PNVP homopolymers was inevitable in the final reaction medium. These impurities were removed by treatment of the crude product with selective solvents, ethyl acetate, to remove PεCL homopolymer and water to remove PNVP homopolymer. Several characterization techniques were employed to study the micellization properties of these amphiphilic block copolymers in aqueous solutions. Rather low hydrodynamic radii values were measured and the cmc values were increased upon increasing the PNVP chain length.

The same reaction series was also adopted for the synthesis of PDLLA-b-PNVP amphiphilic block copolymers (Figure 52) [142]. εCL was replaced by DLLA. PDLLA homopolymers of low molecular weight and moderate dispersities were obtained. The transformation reactions were very efficient and afforded high yields. Good control over the polymerization of NVP was also obtained. However, contamination of the desired products with small amounts of PDLLA and PNVP homopolymers cannot be ruled out. The amphiphilic character of these block copolymers was manifested through the formation of rather small nd spherical micelles in aqueous solutions. The semicrystalline nature of these structures was studied by Differential Scanning Calorimetry, DSC, and Thermogravimetric analysis, TGA.

##### Functional CTAs

Interesting combinations of different blocks with varying properties (solubility in water or in organic solvents, crystalline, amorphous, high Tg or low Tg polymers, etc.) can be prepared employing functional CTAs. These are typical CTAs, capable of promoting RAFT polymerization. In addition, they carry a functional group, e.g., hydroxyl, carboxyl, or amine, either as part of the R or the Z moiety. Therefore, this group remains at the end of the polymer chain after the RAFT polymerization and can be subsequently employed either directly or after suitable chemical transformation as initiator for the growth of another block by a different polymerization technique. The most crucial parameter ensuring the success of the method is the efficient incorporation of the R or Z moieties as end-groups of the first block. Therefore, termination or transfer reactions should be minimized to avoid the presence of unfunctionalized blocks and ultimately the synthesis of ill-defined block copolymers.

This methodology was adopted for the synthesis of the [poly(N-vinyl pyrrolidone)-stat-poly(3-ethyl-1-vinyl-2-pyrrolidone)]-b-poly(ε-caprolactone), [PNVP-stat-PC_2_NVP]-b-PεCL, block terpolymers [58]. For this purpose, S-(1-methyl-4-hydroxyethyl acetate) O-ethyl xanthate, MHEX, was employed as the functional initiator, as shown in Figure 53. Initially, the statistical RAFT copolymerization of NVP and C_2_NVP was conducted in ethanol solution at 70 °C in the presence of MHEX and AIBN. In order to ensure the highest level of functionalization, the polymerization was terminated at rather low conversions (lower than 50%). Measuring the reactivity ratios of the comonomers, it was revealed that both values were close to unity, indicating the synthesis of a copolymer with almost ideal random distribution of monomer units. The end-hydroxyl group of the first block was then used as initiator to promote the ROP of εCL. The reaction was performed at 60 °C in the presence of the organocatalyst diphenyl phosphate. SEC analysis confirmed that both the statistical copolymer and the final block terpolymer have relatively narrow molecular weight distribution and that rather well-defined products were obtained through this procedure. The self-assembly behavior of the terpolymers was studied in aqueous solutions. It was concluded that these products may be effectively applied as drug and gene delivery systems, and also in diagnostic imaging and tissue engineering.

##### Linking Reactions of Individual Blocks

Direct coupling of preformed polymer chains enables the synthesis of block copolymers. This can be accomplished through the interaction of a macroanion and macrocation or through the reaction of specific end-functional groups of the polymer chains. In order to produce well-defined products, special care should be given to choose a quantitative and easy-to-perform coupling reaction. The molecular weights of the individual blocks also play a decisive role in the success of the synthetic methodology. The higher the molecular weight of the chains, the less efficient and the more time consuming is the coupling reaction.

One of the most efficient ways to achieve chain coupling is to perform “click chemistry” reactions [143,144,145]. The 1,3-dipolar cycloaddition reaction between alkynes and azides is by far the most popular click reaction, since it is quantitative, selective and easy to perform. Numerous complex macromolecular architectures have been constructed using this click reaction as the main synthetic tool. Above all, this procedure has also been implemented for the synthesis of block copolymers, where each block has been produced by a different polymerization mechanism.

A combination of RAFT and ATRP techniques was attempted for the synthesis of PNVP-b-PS amphiphilic block copolymers [146]. The procedure involved the click reaction between alkyne terminated PNVP chains from RAFT polymerization and azide-terminated PS chains from ATRP. Specially designed alkyne-containing CTAs X_3_ and X_4_ were synthesized according to the synthetic route presented in Figure 54. Propargyl alcohol was reacted with 2-bromopropionyl bromide to give O-propynyl-2-bromopropionate followed by reaction with potassium O-ethyl xanthate to afford S-2-(propynyl propionate)-(O-ethyl xanthate), X_3_. Along with the desired product (O-ethylcarbonodithionato)-propynyl-(O-ethylcarbonodithionate), X_3′_, was also formed. This byproduct was eliminated by column chromatography. A similar pathway was adopted for the preparation of X_4_.

The polymerization of NVP was performed in bulk at 60 °C using AIBN and either X_3_ or X_4_ as the CTA. Better control was obtained with X_3_. However, even in this case, the monomer conversion was up to 61% and the dispersity values around 1.50. It was found that upon increasing the monomer conversion, gradual loss of the xanthate moieties was observed due to considerable chain transfer reactions. In addition to this, polymers of broader distributions were obtained, and non-symmetrical SEC traces were observed. End-group analysis by NMR techniques and re-initiation experiments with addition of a new amount of NVP confirmed the loss of end groups and the not very well-controlled nature of the polymerization.

Azide-terminated PS was synthesized by ATRP by reaction of the end-bromine group with NaN_3_. The click reaction between the PS and the PNVP chains was performed in DMF solutions at 80 °C using CuBr as the catalyst. Two samples were finally prepared. One of them showed relatively narrow molecular weight distribution, whereas the SEC trace of the other was bimodal indicating that a substantial amount of the PS precursor was left unreacted in the mixture. Poly(D,L-Lactide-co-glycolide)-b-poly(N-vinyl pyrrolidone), PLGA-b-PNVP, block terpolymers were synthesized by ROP, RAFT and click chemistry approaches, as presented in Figure 55 [147]. The RAFT approach was carried out with a functional CTA having an azide functional group, namely S-2-(4-azidobutyl propionate)-(O-ethyl xanthate). The polymerization of NVP was conducted in bulk, at 80 °C, leading to rather low-molecular-weight samples with relatively narrow molecular weight distribution at high conversions. The copolymerization of DLLA and glycolide was performed at room temperature, using propargyl alcohol as the functional initiator and 1,8 diazabicyclo[5.4.0]undec-7-ene, DBU as the catalyst. The polymerization yield was 53.5% and the sample had a low dispersity value. The click reaction took place in DMF solutions at 80 °C using CuBr as the catalyst. SEC and NMR analysis revealed that well defined products were obtained through this approach. Spherical micelles were observed in aqueous solutions and the anti-cancer hydrophobic drug was effectively encapsulated in the core of the micellar structures. In vitro drug-release studies revealed the potential to apply these copolymers in cancer treatment.

A more demanding reaction series was attempted for the synthesis of poly(N-vinyl pyrrolidone)-b-poly(γ-benzyl-L-glutamate), PNVP-b-PBLG, block copolymers [148]. It is based on the reaction of an aldehyde with 1,2-aminothiol leading to the formation of the stable and biocompatible thiazolidine linkage, as described in Figure 56. This reaction takes place without the need of any external stimuli or catalyst. The conjugation reaction for the synthesis of the desired amphiphilic block copolymer is displayed in Figure 57 and involves the reaction of an aldehyde terminated PNVP chain with an 1,2-aminothiol terminated PBLG chain.

NVP was polymerized employing a typical RAFT procedure. The reaction took place in bulk at 60 °C with AIBN as the radical source and S-(1-((2-(1H-indol-3-yl)ethyl)amino)-1-oxopropan-2-yl) O-ethyl carbonodithioate as the CTA. A relatively low-molecular-weight (less than 10,000) sample with a moderate polydispersity (Ð = 1.32) was obtained at 52% conversion. The xanthate moiety was hydrolyzed in aqueous acidic solution at 40 °C. The end-hydroxyl group which was formed was almost quantitatively transformed to the aldehyde end-group by thermolysis at 120 °C under vacuum (Figure 58). NMR and MALDI-TOF techniques were used to verify the success of this treatment.

PBLG was obtained typically by the ROP of the corresponding N-carboxy anhydride, NCA, at 0 °C in DMF solution, using benzylamine as the initiator. It was known that the end γ-benzyl-L-glutamate unit undergoes a cyclization reaction, forming pyroglutamate and thus leading to the loss of the end amine group. In order to avoid this, the PBLG chains were end-capped with a single cysteine moiety. The amine group of this cysteine unit was protected with an Fmoc group, whereas the thiol group was protected with an acetamido group, Acm. After conjugation of the cysteine unit with the PBLG chain both protecting groups, Fmoc and Acm were sequentially removed. The deprotection step of the Acm group is very tedious and enables the danger of hydrolysis of the main polymer chain. However, the protocol that was adopted resulted in efficient deprotection and oxidation of the thiol functionality and finally to the desired product with controlled molecular characteristics (Figure 59).

The conjugation of the aldehyde functionalized PNVP and the cysteine functionalized PBLG took place using equimolar ratios of the two polymers in DMF solution in the presence of DL-dithiothreitol. SEC analysis revealed that the final products have much broader molecular weight distributions and that the copolymer traces have a partial overlap with those of the constituent blocks, meaning that small amounts of these blocks remained unreacted during the conjugation. Dynamic light scattering, DLS, and transmission electron microscopy, TEM, were mainly employed to study the self-assembly behavior of the copolymers. It was found that the micellar structures had excellent cell compatibility, even at high concentrations, thus making this system efficient for drug delivery applications.

#### 4.2.4. Triblock Copolymers and Terpolymers Based on NVP Synthesized via RAFT Polymerization

Triblock copolymers or terpolymers make up a very interesting class of polymeric materials with unique properties both in solution and in bulk [149]. The incompatibility between the different blocks leads to self-assembled nanostructures in selective solvents and microphase separation in bulk. The triblocks can be divided into the following categories:(a)Symmetric triblock copolymers of the type A-b-B-b-A, where the A blocks have the same molecular weight.(b)Asymmetric triblock copolymers of the type A-b-B-b-A’, where the A and A’ blocks are chemically identical but have different molecular weight.(c)Triblock terpolymers of the type A-b-B-b-C.

The synthesis of the triblocks involving RAFT polymerization can be accomplished through various methodologies [88]:(a)Use of two CTAs chemically connected through the Z group.(b)Use of two CTAs chemically connected through the R group.(c)Use of double CTAs, e.g., CTAs with two leaving groups Z on the same molecule.(d)Sequential addition of monomers using the same monofunctional CTA.(e)End group functionalization of a polymer prepared by a non-RAFT methodology in order to incorporate a suitable CTA moiety, followed by RAFT polymerization of one or two suitable monomers.(f)Use of a functional CTA able to promote RAFT and another type of polymerization.

These approaches are schematically described in Figure 60. Methods 1–3 can lead to symmetric A-b-B-b-A triblock copolymers, whereas methods 4-6 can produce A-b-B-b-A’ asymmetric triblock copolymers or A-b-B-b-C triblock terpolymers.

These methodologies can be applied for the synthesis of PNVP-based triblocks as will be described in the next sections.

A double CTA containing two leaving Z groups, namely S,S’-bis(α,α’-dimethyl-α’’-acetic acid)-trithiocarbonate, was employed for the synthesis of poly(styrene-co-acrylic acid)-b-poly(N-vinyl pyrrolidone)-b- poly(styrene-co-acrylic acid), P(S-co-AA)-b-PNVP-b-P(S-co-AA) triblock terpolymers [150]. The reaction sequence is given in Figure 61. The statistical copolymerization of S and AA was initially performed in DMF using the double CTA and AIBN. The copolymerization was conducted at 80 °C. The double macro-CTA that was synthesized was further employed for the polymerization of NVP in DMF at 80 °C using 4,4′-azo-bis(4-cyanovaleric acid) as the radical source. The authors reported the presence of PNVP homopolymer in the crude product, which was eliminated by treatment with selective solvents. Only one sample was presented without detailed molecular characterization. However, this sample was employed to improve the blood compatibility of polyethersulfone, PES, membrane surfaces. The triblock was directly blended with the PES matrix leading to a modified membrane with superior properties compared to the unmodified membrane. Consequently, these products can be efficiently used in blood purification and hemodialysis.

The same CTA was also employed for the synthesis of PNVP-b-PMMA-b-PNVP triblock copolymers, as shown in Figure 62 [151,152]. In this case, NVP was polymerized first in aqueous solution at 80 °C using 4,4′-azo-bis(4-cyanovaleric acid), ACVA, as initiator. This bidirectional CTA was further employed for the polymerization of MMA in DMF at 80 °C with AIBN. The characterization in this case was also incomplete, since detailed SEC and NMR analysis is missing. However, the authors reported the contamination of the product with diblocks and homopolymers, indicating that the reaction scheme leads to samples with pronounced chemical heterogeneity. In another work, the same group provided characterization data of the triblock copolymers showing that the polymerization conversions were not very high and that the products had rather broad molecular weight distributions. These products were also blended with PES matrix to modify the original membranes to improve the blood compatibility. The final materials showed good ultrafiltration and protein anti-fouling properties and improved cytocompatibility.

A CTA combining two other CTAs chemically linked through the Z group was synthesized and employed for the synthesis of poly(N-isopropylacrylamide)-b-PNVP-b-poly(N-isopropylacrylamide), PNIPAAm-b-PNVP-b-PNIPAAm, triblock copolymer, according to Figure 63 [153]. The CTA 1,4-phenylenebis(methylene) bis(ethyl xanthate) was prepared by reaction of 1,4-bis(bromomethyl)benzene with potassium O-ethyl xanthate and was employed for the polymerization of NVP in 1,4-dioxane at 70 °C with AIBN as initiator. This macro-CTA was subsequently used for the bidirectional polymerization of NIPAAm under the same experimental conditions. SEC and NMR analysis revealed that very good control was achieved over the polymerization of both monomers and that relatively pure products were obtained. These triblocks were further reacted with NIPAAm and the difunctional monomer N,N’-methylenebisacrylamide in the presence of AIBN to provide thermoresponsive hydrogels.

The same CTA was also used for the synthesis of PNVP-b-PVK-b-PNVP triblock copolymers, where PVK is poly(N-vinyl carbazole) by RAFT polymerization and sequential addition of monomers (Figure 64) [154]. VK was polymerized first in 1,4-dioxane at 60 °C with AIBN and the difunctional CTA to products of very high yields and narrow molecular weight distribution. Subsequent addition of NVP in 1,4-dioxane at 70 °C with AIBN and the macro-RAFT agent led to the synthesis of the desired product. The polymerization yield was very high again and the reaction was very well controlled, leading to low chemical heterogeneity. More samples with detailed characterization data confirming the previously mentioned conclusions were prepared and studied in another publication [155]. TEM analysis of the triblocks revealed that they are microphase separated. In addition, they self-assemble in aqueous solutions forming spherical micelles, having sizes that depend on the length of the PNVP blocks. The same structures were incorporated into epoxy resins leading to the formation of nanostructured thermosets containing PVK nanophases. Both the photoluminescent and dielectric properties were improved compared to the unmodified epoxy resins.

Commercially available PDMS with rather broad molecular weight distribution bearing hydroxypropyl groups at both chain ends was transformed to bifunctional macro-CTA after reaction with bromopropionylbromide followed by reaction with potassium O-ethyl xanthate (Figure 65) [156]. Subsequent polymerization of NVP in THF at 60 °C with AIBN afforded the PNVP-b-PDMS-b-PNVP triblock copolymers. Rather well-defined low-molecular-weight triblocks were obtained, as revealed by SEC analysis. Initial studies revealed that these triblocks were highly surface active in aqueous media and can self-assemble in bulk providing spherical PDMS structures.

ATRP and RAFT methodologies were combined for the synthesis of poly(methyl acrylate)-b-poly[(7-(allyloxy)-2H-chromen-2-one)-co-(2-hydroxyethyl methacrylate)]-b-poly(N-vinyl pyrrolidone), PMA-b-P(AC-co-HEMA)-b-PNVP triblock quaterpolymers, as shown in Figure 66 [157]. ATRP was initially performed for the synthesis of the PMA block using CuBr as the catalyst, N,N,N′,N″,N″-pentamethyldiethylenetriamine, PMDETA, as the ligand and ethyl α-bromoisobutyrate, EBiB, as the initiator. The reaction took place in THF at 65 °C up to 80% yield. Subsequent addition of HEMA and AC in the presence of PMDETA and CuBr in THF solution provided the second block. The polymerization was conducted at 65 °C with a yield up to 30%. The end-bromine group of the diblock terpolymer was then reacted with potassium O-ethyl xanthate to provide the corresponding diblock macro-CTA, which is capable of polymerizing NVP via the RAFT methodology. This final reaction step was carried out in THF at 60 °C using AIBN. The polymerization yield was up to 70%. Products of low molecular weights were obtained in almost all cases. SEC analysis revealed the presence of monomodal peaks of low dispersity for the final triblocks without obvious termination or other side reactions. These terpolymers were found to self-assemble to spherical micelles in aqueous solutions with PMA cores and PNVP coronas. A secondary aggregation process was obtained above the critical micelle temperature, leading to the formation of cubic morphologies. Curcumin was effectivelly encapsuated in these supramolecular structures and the release profiles highly depend on the temperature. Photo-crosslinking of the micelles was also accomplished by UV-irradiation, through the photosensitive middle block of the triblocks.

### 4.3. Star Polymers

#### 4.3.1. Introduction

Star polymers make up the simplest sub-group of branched polymers. They are constructed from linear chains linked to a central core [158]. This core may be a single atom, a small molecule or even a macromolecular structure. In all cases, the size of the core should be much smaller than the overall size of the star branched macromolecule. The high segment density of these polymers results in the formation of compact structures, compared to their linear counterparts, thus leading to unique solution, viscoelastic and mechanical properties.

In polymer chemistry, the synthesis of star polymers has been the subject of all the available polymerization techniques [159,160,161,162,163,164,165]. Combinations of various techniques have been also reported in the literature for the synthesis of special categories of star polymers. Several star architectures have been exploited in the past including regular stars of different functionalities, star-block copolymers, functionalized stars, asymmetric stars and miktoarm stars, as depicted in Figure 67.

Two major and different venues, the arm-first and the core-first techniques, have been employed during the years for the synthesis of star polymers, as shown in Figure 68. The arm-first, or arm-in, or convergent approach involves in the first step the synthesis of living polymeric chains and then in a second step, their subsequent linking to a multifunctional linking agent. This process involves many advantages, since the characterization of the product is straightforward monitoring the synthesis by SEC and measuring the molecular weights of both the arms and the star-branched polymers. The functionality of the linking agent defines the number of the arms provided that the linking reaction is quantitative. The main disadvantages of this approach are the long linking reaction times, the steric hindrance problems encountered with the linking process and the need to use an excess of the living arm to assure complete linking. The excess of the arm should be later eliminated by a suitable purification method, such as fractionation or dialysis.

The core-first or arm-out or divergent approach involves the employment of multifunctional compounds that are capable of simultaneously initiating the growth of several arms. The main requirement for the efficient application of the method is that all initiating sites should be equally reactive, with the initiation rate being much higher than the propagation rate. Under these conditions, stars with equal arm molecular weight and low molecular and chemical heterogeneity can be obtained. In this case, the arm molecular weight cannot be directly measured and therefore, the number of arms can be calculated by end-group analysis or by measuring typical branching parameters, such as the ratios of mean square radius of gyration, intrinsic viscosity and hydrodynamic radius of the star polymer over the value of corresponding linear polymer having the same molecular weight as the star.In RAFT polymerization, the core-first methodology was further partitioned into two separate processes [88]. Over the years, the core-first method has been thoroughly exploited, though later there was a shift of focus on the arm-first technique. This can be attributed to the increasing interest in end-group chemistry, since the RAFT protocol might be combined with other pathways such as click chemistry or amide bond formation to produce star-shaped polymers through the arm-first venue.

#### 4.3.2. Star Polymers following the Core-First Technique

As already mentioned, RAFT polymerization involves two different approaches of the core-first strategy, both depending on the chain transfer agent (CTA). The general form of these thiocarbonylthio species that serve as RAFT agents is depicted in Figure 69 [88]. The Z and R groups of the RAFT agent determine its reactivity along with its compatibility with various monomers and the unique properties to the agent that are later found in the polymer. The structural features of the Z group control the reactivity of the CTA by affecting the reactivity of the C=S bond and its ability towards radical addition. On the other hand, the structure of the R group affects the overall reactivity of the agent and has significant influence over the polymerization kinetics, including the overall degree of control. However, under no circumstances can it be assumed that these two entities act irrespectively to one another. To find an effective combination of the Z and R groups can be a precarious process, which does not always lead to an efficient RAFT agent. The core-first technique steps on the potential provided by the chain transfer agent chemistry and introduces two different synthetic routes, the Z-group and the R-group approach, which depend on the Z and R moieties, respectively. Following the core-first process, stars with a number of arms that equals the number of chain transfer sites on the core can be synthesized. The RAFT agent can be bound to the core from either the R or the Z group, which will convey the previously barren core to a multifunctional CTA. To decide which part of the RAFT agent should be bound to the core is not a decision to be taken lightly, since it can impinge upon the outcome of the process to be discussed below.

##### R-Group Approach

As in traditional RAFT synthesis, the R-group approach polymerizations need a radical source to introduce initiation. In this scenario, the CTA is connected to the core of the star through the R-group. Following the steps of a regular RAFT polymerization, the radical upon its creation is added forcefully to the RAFT agent. After that, the dithioester part detaches from the core and initiates polymerization, while the core becomes a radical itself. With the R-group approach, it is understandable that a considerable number of terminating reactions can occur, since there are more radicals in the polymerization solution. With the resulting increasing number of arms on the multifunctional RAFT agent, the bigger the chances of star-star coupling termination reactions. An increased probability for the formation of by-products results in a broad molecular weight distribution, leading to multi-modal distributions. A mechanistic approach of the R-group methodology is depicted in Figure 70.

According to the Barner–Kowollik computational modeling program, a sufficient number of experimental conditions have been predicted to help us ensure the appropriate conditions for a prosperous R-group approach venue [166,167]. The factors to be considered are (a) the initiator concentration, which in agreement with the conventional RAFT process should be kept low in comparison to the CTA concentration, (b) the number of arms of the multifunctional chain transfer agents, since an increasing number of arms results in an increasing number of radicals in the solution, facilitating an increase in star–star coupling events, and (c) the monomer reactivity ratio, i.e., monomers with a higher reactivity ratio can achieve higher conversions in a shorter period of time, and therefore a smaller number of radicals is necessary for the process, therefore reducing the chance of side reactions.

##### Z-Group Approach

On the other side of the core-first method coin lies the Z-group approach. As stated previously, during this route, the RAFT agent is permanently tied to the nucleus, while the polymerization takes places in the solution around it (Figure 71). Since the chain transfer agent does not detach from the core, there is no presence of radicals, so it appears to be impossible to have star–star coupling reactions. Therefore, in this case only a small number of termination reactions are present, leading to the formation of linear byproducts (Figure 72).

During a Z-type approach, steric hindrance has been observed, which does not allow for the synthesis of high-molecular-weight polymers or stars with a high number of arms. In experiments using styrene, even though a monomodal distribution was monitored, there was a large deviation between theoretical and experimental molecular weight. This was due to the delay in growth of the polymer arms caused by the shielding effect of the arms, where during polymerization each arm grows from the thiocarbonylthio groups attached to the core; hence, the polymer chains surrounding the core act as a shield preventing further chain transfer. The shielding effect in this case is responsible for the increased number of termination reactions.

It is not possible to reach a general conclusion as to which of the two techniques is to be considered superior, since each method has advantages. It is clear that both pathways can deviate from the main product through multiple side reactions which cannot be completely repressed but only reduced. The monomer itself can indicate which of the methods should be used, as the propagating pace of the monomer can be of substantial importance in the broadening of the distribution as well as the termination reactions that are to occur. Careful design of all the parameters along with meticulous research prior to the choice of the R- or the Z-group approach can lead to good results.

Although both approaches have been efficiently employed for the synthesis of star polymers, only the R approach has been applied in the case of PNVP stars. Following this methodology, four-arm star PNVP homopolymers were synthesized. A suitable tetrafunctional CTA was synthesized following the route given in Figure 73 [168]. 1,2,4,5 Tetrakis[(O-ethylxanthyl)methyl]benzene was prepared by reaction of tetrakis(bromomethyl)benzene and O-ethyl xanthic acid potassium salt. The polymerization of NVP was performed in bulk at 60 °C using this product as CTA and AIBN as the initiator (Figure 74). The molecular weight of the star polymer increased linearly with conversion up to 70% conversion. Until then, relatively narrow molecular weight distributions were obtained. At higher conversions, broader distributions and deviations from the linearity between molecular weight and conversion were noted, a case attributed to the typical side reactions of this technique. The absence of typical byproducts of the R-group approach up to conversion equal to 70% can be attributed to low radical concentrations and the fast propagation of the monomer. The star polymers were employed as stabilizers in suspension polymerization for the preparation of crosslinked poly(vinyl neodecanoate)ethylene glycol dimethacrylate microspheres.

The tetrafunctional RAFT agent 2,2′-oxybis(methylene)bis(2-ethylpropane-3,2,1-triyl)tetrakis[2-(ethoxy carbonothioylthio)propanoate] was synthesized as shown in Figure 75 [169]. Di(trimethylolpropane) was initially reacted with 2-bromopropionyl bromide to afford 2-bromo-propionic acid 2-[2,2-bis-(2-bromo-propionyloxymethyl)-butoxymethyl]-2-[2-bromo-propionyloxymethyl)-butyl ester intermediate, which was then reacted with potassium O-ethyl xanthate to give the desired product. The polymerizations with this RAFT agent were conducted in 1,4-dioxane at 70 °C using ACVA as initiator. Homopolymerization of NVP gave the symmetric 4-arm PNVP stars. Simultaneous copolymerization of NVP with VAc produced the statistical star branched copolymers (PVAc-stat-PNVP)_4_ (Figure 76). Sequential polymerization of VAc first followed by the polymerization of NVP in a subsequent step resulted in the synthesis of (PVAc-b-PNVP)_4_ 4-arm star block copolymers (Figure 77). SEC and NMR analysis revealed the presence of rather well-defined star polymers. In the case of the star block copolymer, the dispersity of the first homo-star increased substantially after the polymerization of the second monomer, indicating a small loss of control of the polymerization. This was obvious from the SEC traces of the four-arm PVAc initial star and the final star block copolymer. The first trace was symmetric, while the second one was bimodal.

RAFT polymerization under high pressure was attempted in an effort to minimize star–star and star–chain coupling side reactions, which were reported to take place during the R-group approach for the synthesis of star polymers. This methodology was applied even at quantitative polymerization conversions. For this purpose, the CTA pentaerythritol tetrakis [2-(dodecylthiocarbonothioylthio)-2-methylpropionate] was synthesized and employed for the synthesis of four-arm PNVP stars [170]. The polymerization was conducted in bulk at 60 °C under a pressure of 250 MPa (Figure 78). Star-shaped polymers with a wide range of molecular weights and relatively low dispersities were obtained in almost quantitative conversion. Comparison with similar experiments conducted at ambient pressure was provided, manifesting the critical role of employing high pressure as a tool to suppress the side reactions that are accommodated with the R-group strategy. The four-arm PNVP stars were also used to promote the photo-induced RAFT polymerization of MMA under UV irradiation at 365 nm, leading finally to the synthesis of star-block amphiphilic copolymers.

Six-arm star copolymers and terpolymers were obtained employing a hexafunctional CTA derived from dipentaerythritol, which was prepared following similar procedures, as described previously [168]. The statistical star copolymers (PNVCL-co-PNVP)_6_ and the block-statistical terpolymers [(PNVCL-co-PNVP)-b-PVAc]_6_ and [PNVAc-b-(PVCL-co-PNVP)]_6_ have been prepared (NVCL stands for N-vinylcaprolactam), as shown in Figure 79 [171]. The statistical stars were synthesized by the copolymerization of the two monomers at 30 °C in 1,4-dioxane using 2,2′-azobis(4-methoxy-2,4-dimethyl valeronitrile) as initiator in the presence of the hexafunctional CTA. For the synthesis of the [(PNVCL-co-PNVP)-b-PVAc]_6_ terpolymer star, the statistical copolymer (PNVCL-co-PNVP)_6_ was employed as the macro-CTA. The polymerization of VAc was conducted at 80 °C in 1,4-dioxane using 4,4-azobis(4-cyanovaleric acid) as the radical source. Finally, for the synthesis of the [PNVAc-b-(PVCL-co-PNVP)]_6_ terpolymer star, the six-arm (PVAc)_6_ star was initially prepared followed by the copolymerization of NVCL and NVP, under identical conditions as previously reported for the other terpolymer star, except that the copolymerization temperature was 65 °C. NMR and SEC analysis revealed the synthesis of well-defined star structures in all cases. The star block copolymers were found to form thermoresponsive flower-like micelles in aqueous solutions. These micelles were employed as vehicles to efficiently encapsulate and release in a controlled fashion of methotrexate.

A combination of ROP and RAFT polymerization techniques was employed for the synthesis of (Pε-CL-b-PNVP)_4_ four-arm star block copolymers [172]. Pentareythritol, in the presence of Sn(Oct)_2_, was employed as initiator to provide the corresponding (Pε-CL)_4_ homopolymer star. The end-OH groups at each arm were then reacted with 2-bromopropionyl bromide to provide the corresponding end-Br groups. These groups were finally converted to O-ethyl xanthate groups upon reaction with potassium O-ethyl xanthate. This product served as tetrafunctional CTA during the polymerization of NVP, leading to the synthesis of the star block copolymers. The final reaction took place in THF at 80 °C, using AIBN as initiator (Figure 80). In order to avoid the side reactions, the conversion of NVP was kept low (lower than 50%). Under these conditions, rather well-controlled star structures were obtained. The amphiphilic star block copolymers formed spherical micelles in aqueous solutions and were further employed as scaffolds for the synthesis of silver nanoparticles.

The exact same procedure was adopted for the synthesis of amphiphilic (PDLLA-b-PNVP)_4_ star block copolymers, as shown in Figure 81 [173]. As reported in the previous case, the NVP conversion was kept low, actually much lower (up to 21.3%), in order to avoid side reactions and obtain products with minimum molecular and structural heterogeneity. This was verified by NMR and SEC analysis of the intermediate and the final products. Spherical micelles were obtained in aqueous solutions. These micelles were able to efficiently encapsulate methotrexate and thus to show significant growth inhibition, cytotoxicity and apoptosis of certain types of cells. The amphiphilic star block copolymers also exhibited antitumor activity.

The same combination of ROP and RAFT techniques was employed for the synthesis of amphiphilic (Pε-CL-b-PNVP)_3_ three arm star block copolymers [174]. Triethanolamine, in the presence of dibutyltin dilaurate was employed as initiator for the polymerization of ε-CL at 140 °C and the formation the three-arm (Pε-CL)_3_ star. NVP was polymerized independently, by RAFT in 1,4-dioxane solutions at 80 °C, using thioglycolic acid as CTA. The linear PNVP chains with the end-COOH groups were then linked to the arms of the (Pε-CL)_3_ star through esterification reaction, employing DCC and DMAP, as shown in Figure 82. Although the SEC data looked promising, the corresponding traces were not provided to assess whether these were symmetrical or not or if there was an excess of the PNVP chains after the coupling reaction, etc. Spherical micelles were found to be formed in aqueous media. The encapsulation of folic acid and its efficient release upon changing the solution pH reveal that these micelles can be efficiently employed in drug delivery applications.

#### 4.3.3. Star Polymers Following the Arm-First Technique

The arm-first strategy begins with pre-synthesizing a branch, regarded as a macro-RAFT agent, which is later conjugated to a multifunctional core containing groups that will chemically react with the end groups on the polymer chains, therefore affording a star polymer. In the course of this process, good knowledge of organic chemistry paves the way for a successful synthesis.

Along these lines, the synthesis of seven-arm PNVP stars with a protein core, in particular lysozyme, has been reported in the literature. O-ethyl-S-(p-methyl benzoylsuccinimide) xanthate was employed as CTA for the controlled RAFT polymerization of NVP leading to N-succinimidyl ester terminated polymers [175]. The polymerization was conducted in bulk at 60 °C using AIBN as the radical source. Lysozyme-containing six lysine residues carries seven primary amine groups, six from the lysine units and the terminal amine group. These amine groups may react with the N-succinimidyl ester functions of the PNVP chains, thus leading through this coupling reaction to the synthesis of seven-arm star polymers. The reaction series is given in Figure 83. The polymerization of NVP with the functional CTA was well controlled producing polymers with relatively low dispersity values. Only at higher conversions was the dispersity increased due to the occurrence of termination reactions. Detailed analysis, by NMR, MALDI or any other characterization technique, of the quantitative presence of the end-group was not reported. The linking reaction between lysozyme and the end-functionalized PNVP was conducted in DMSO in the presence of triethylamine and using a large excess of the polymeric arms in order to ensure complete coupling. The reaction was monitored by SEC. The analysis showed that when rather low-molecular-weight arms (less than 20,000) were employed, well-defined products of narrow molecular weight distribution can be obtained. However, upon using higher-molecular-weight PNVP chains, several byproducts were observed, i.e., star structures with functionality less than 7. Extensive purification of the star polymer and careful molecular weight characterization is still needed to unambiguously verify the impact of this methodology as a general arm-first method for the synthesis of PNVP stars.

### 4.4. Graft Copolymers

#### Introduction

Graft copolymers are composed of a main linear polymeric chain, the backbone, to which one or more side chains, or branches, are chemically connected through covalent bonds [159,160,161,164,176,177]. The backbone and the branches may be homopolymers or copolymers. When both the backbone and the branches have the same chemical nature and composition, the branched structures are characterized as combs, whereas when they differ in chemical nature or composition, they are called grafts. In the present work the more general term graft copolymer will be used. The branches are usually equal in length and randomly distributed along the backbone because of the specific synthetic techniques employed for their preparation. However, more elaborate methods have been developed for the synthesis of regular graft copolymers with equally spaced and identical branches and of exact graft copolymers, where all the molecular and structural parameters can be accurately controlled (Figure 84).

Three general methods have been developed for the synthesis of randomly branched graft copolymers: (1) the “grafting onto”, (2) the “grafting from” and (3) the macromonomer method (or “grafting through” method) (8) (Figure 85).

The “grafting onto” method involves the use of a polymeric backbone-containing functional groups X randomly distributed along the chain and branches having reactive chain ends Y. The coupling reaction between the functional backbone and the end-reactive branches lead to the formation of graft copolymers.

The characterization of the backbone and the preformed side chains can be performed separately from the graft copolymer, thus allowing for the detailed characterization of the final structure. If a living/controlled polymerization is employed, well-defined backbone and branches can be prepared and therefore the graft copolymer will have the maximum degree of structural control. Disadvantages of this technique can be considered the following: (a) high grafting density is prohibited due to the increased steric hindrance effects during the linking reaction among the backbone and the branches. A direct consequence of this fact is that polymer brushes cannot be synthesized via this methodology. (b) Rather low-molecular-weight branches are usually employed to afford complete linking.

In the “grafting from” method, active sites are generated randomly along the backbone. These sites are capable of initiating the polymerization of a second monomer, leading to graft copolymers.

With this approach it is easy to control the molecular characteristics of the backbone, provided that a living/controlled polymerization technique is employed. In addition, high grafting density can be achieved leading even to the efficient synthesis of polymer brushes.

The number of grafted chains can be controlled by the number of active sites generated along the backbone assuming that each one participates in the formation of one branch. Full structural characterization of the products obtained from a grafting from scheme is very difficult since neither the exact number of side chains added, nor their molecular weight can be determined. Usually, the branches have a broad molecular weight distribution and cannot be isolated and separately characterized.

The most commonly used method for the synthesis of graft copolymers is the macromonomer method [177]. Macromonomers are oligomeric or polymeric chains bearing a polymerizable end group. Macromonomers with two polymerizable end groups have also been reported. Copolymerization of preformed macromonomers with another monomer yields graft copolymers.

It is possible through this technique to synthesize polymer brushes or at least copolymers with high grafting density. Depending on the graft length and degree of polymerization the polymacromonomers may adopt several conformations in solution, such as star-like, comb-like, bottle-brush or flower-like. The molecular characteristics of the branches can be easily controlled. The synthesis of macromonomers can be accomplished by almost all the available polymerization techniques. Among these techniques, living polymerization methods offer unique control over molecular weight, molecular weight distribution and chain-end functionalization.

The drawbacks of this approach are the employment of rather low-molecular-weight branches and the fact that the backbone cannot be isolated for detailed characterization.

The general methodologies for the synthesis of graft copolymers may be employed with pure RAFT polymerization or with combinations with other polymerization techniques taking into account the specific features of the RAFT technique [88]. Copper catalyzed azide-alkyne cycloaddition (click reaction) was employed as the tool to link alkyne-terminated PNVP chains with hydroxyethyl cellulose, HEC, which was modified to carry azide groups in a typical grafting “onto” methodology (Figure 86) [178]. The alkyne-terminated PNVP was prepared employing O-ethyl-S-propyl-2-ynyl-carbonodithiolate as the CTA. The polymerization was conducted in toluene at 70 °C using AIBN as the radical source. A very low-molecular-weight sample (M_n_ < 2000) was obtained in order to facilitate the spectroscopic characterization. The conversion was as high as 80%, and the product had a reasonably low dispersity (Ð = 1.4). The functionalization of HEC was performed by reaction with NaN_3_ in DMF solutions in the presence of triphenylphosphine at room temperature. Due to the insolubility of the azide-functionalized HEC in most organic solvents, the characterization of the product was performed in the solid state mainly by FT-IR and NMR techniques (^13^C and ^15^N-CP-MAS solid state NMR spectra). The click reaction took place in DMF solutions at 30 °C for 24 h in the presence of copper (II) sulphate, sodium L-ascorbate, and N,N,N’,N’-tetramethylethylenediamine. The products were characterized again by FT-IR and NMR techniques.

The grafting “from” approaches are more commonly employed for the synthesis of graft copolymers via RAFT [88]. These involve:The attachment of the RAFT agent to the backbone. This approach adopts the main synthetic paths that are followed in its star equivalent synthesis. The CTA can be attached to the backbone either from the Z (Z-group approach) or the R group (R-group approach), following in each case the same mechanistic course with the advantages and disadvantages of each method (Figure 87). In the R approach for the synthesis of graft copolymers, the main difference in the graft synthesis is the concentration of the CTA on the backbone, which is considerably higher in the case of graft monomers than in stars. This leads to a greater occurrence of side reactions, especially graft–graft coupling. In graft polymers, there is a vastly greater number of branches than in star polymers, resulting in increased termination reactions. Therefore, the amount of graft–graft coupling increases upon increasing the number of side chains. Thus, the amount of side products has an immediate effect on the molecular weight distribution of the final product, which can be controlled through various pathways, including (a) keeping the molecular weight of the side chains as low as possible, (b) reducing the concentration of the radicals in the polymerization reaction and (c) lowering the reaction temperature. Similar drawbacks are faced by the Z-group approach for the synthesis of graft copolymers. The most important is the steric shielding effect leading to pronounced termination reactions of two linear macro-radicals.The attachment of radical initiator fragments to the backbone (Figure 88). Important parameters in this case are the amount of the CTA over the initiation sites, which controls the molecular weight of the side chains and the rate of activation of the initiator. The most commonly observed byproducts in this case are linear macro-RAFT agents generated from the leaving R group of the CTA.

PVAc-g-PNVP graft copolymers were synthesized by RAFT copolymerization techniques, the grafting “from” methodology and the R-approach (Figure 89) [179]. VAc and vinyl chloroacetate, VClAc, were copolymerized via RAFT in ethyl acetate at 80 °C employing O-ethyl S-(1-methoxycarbonyl) ethyl xanthate (Rhodixan A1) as the CTA, and 1,1′-azobis(cyclohexane carbonitrile) (V-40) as the radical source. Since rather low grafting density was chosen, the composition of the copolymers in VClAc was rather low (up to 26%). Low-molecular-weight statistical copolymers were prepared with rather broad molecular weight distribution (Ð values ranging from 1.69 to 1.77). It is important to note that the distribution of the pendant chlorine groups along the chain is highly random, judging from the reactivity ratios of VAc and VClAc. The terminal xanthate group of the statistical copolymer was removed via a radical-induced reduction, by treatment of the copolymer with excess of the radical initiator lauryl peroxide and heating at 80 °C. This treatment caused no further change in the molecular characteristics of the copolymer. UV-vis spectroscopy revealed the efficient removal of the end group. The pendant chlorine groups of the copolymers were transformed to a multi-CTA agent upon reaction with potassium ethyl xanthogenate in acetone solution at room temperature. NMR spectroscopy confirmed that up to 50% of the available chlorine groups was transformed to xantahtes. Finally, the R-approach was employed and the grafting from methodology with the polymerization of NVP from the CTA positions along the PVAc-based backbone. The polymerization was conducted in methanol at 60 °C using AIBN as the radical initiator. SEC analysis showed that a very small amount of linear PNVP was formed and that the desired graft copolymers were efficiently produced, having moderate to high dispersity values (Ð values ranging from 1.55 to 1.92). The molecular weight distribution showed a significant increase, especially in PNVP-rich samples. This could be partly attributed to the high conversion of the NVP in those samples. Star-like micelles were observed after the self-assembly of the graft copolymers in aqueous media, thus creating the possibility to be applied in biomedical and cosmetic formulations.

A combination of ROP and RAFT polymerization techniques, the grafting “from” methodology and the R-group approach were employed for the synthesis of PCL-g-(PNVCL-co-PNVP) graft terpolymer, as reported in Figure 90 [180]. The backbone was prepared by ROP of ε-caprolactone, CL, and α-chloro-ε-caprolactone, ClCL, which was promoted by camphorosulfonic acid as the catalyst in the presence of 2,2-dimethyl-1,3-propanediol, DMP, as the difunctional initiator. The reaction was conducted in toluene solutions at 60 °C. It was reported that ClCL polymerizes much faster than CL. Therefore, the statistical copolymerization of these monomers would afford a gradient copolymer with large blocks of either ClCL or CL. This would cause pronounced steric hindrance effects in the subsequent grafting from step, thus leading to low grafting densities. In order to avoid this drawback, the semi-batch approach, i.e., the slow monomer addition method, was adopted. In addition, this approach results in copolymers with lower dispersity values. The pendant chlorine groups were then reacted with potassium ethyl xanthogenate to incorporate CTA moieties along the polymer chain, suitable to promote the RAFT polymerization to monomers belonging to LAMs. This reaction took place in THF solutions at room temperature and was monitored by NMR spectroscopy and SEC, revealing that the substitution step was efficient and no side effects (degradation, crosslinking, etc.) were traced to the polymeric chain. The final step involved the RAFT copolymerization of NVCL with NVP in toluene solution at 40 °C using 2,2′-azobis(4-methoxy-2,4-dimethylvaleronitrile), V-70 as initiator. The products were characterized by NMR spectroscopy. However, detailed molecular characterization of the graft terpolymers was missing. Judging from the reactivity ratios of NVCL and NVP, it can be concluded that the side chains are gradient copolymers. An indication for the successful synthesis of the desired structures could be the LCST (lower critical solution temperature) rise in the polymers when the composition of PVNP varied between 10–20%. The 32 °C LCST of PCL-g-PNVCL was increased to 38–40 °C when PNVP was introduced at the side chain.

As PNVP shows increased biocompatibility and extensive solubility in aqueous solutions, indication that it can be employed in gene delivery applications is given through the synthesis of PNVP-g-PDMAEMA graft copolymers [116]. These materials were studied as non-viral vectors for gene delivery. The process which was followed for the synthesis of the graft PNVP-g-PDMAEMA copolymers involved the polymerization of NVP via RAFT in DMF solution at 60 °C using AIBN as the radical source and S-(2-ethyl propionate)-O-ethyl xanthate as the CTA. The PNVP homopolymer was then subjected to bromination reaction in CCl_4_ solution in the presence of AIBN and N-bromosuccinimide, NBS, under heating at 90 °C. Almost 16% bromination was achieved, as revealed by NMR spectroscopy. The pendant bromine groups were further employed as initiation sites to promote the ATRP of DMAEMA in a typical grafting “from” procedure. The reaction was conducted in DMF at 40 °C using 2,2′-bipyridine as the ligand and CuBr as the catalyst (Figure 91). Through ^1^H-NMR spectra, the successful grafting was verified. SEC traces monitoring the synthetic procedure were not provided. However, only one sample of rather low molecular weight was reported with an extremely low dispersity value (Ð = 1.009), which is very difficult to accept taking into account the applied polymerization techniques and the complex structure that was prepared. Compared to the corresponding block copolymers, the graft condenses DNA more effectively into polyplexes with smaller size, higher zeta potential and better stability.

Bottle-brushes were synthesized by a combination of Atom Transfer Radical Polyaddition, ATRPA, and RAFT. ATRPA was performed to 4-vinylbenzyl 2-bromo-2-phenylacetate, VBBPA [181]. The reaction was utilized in anisole solution at 40 °C in the presence of CuBr_2_/Cu and 4,4′-dinonyl-2,2′-bipyridine, dNbpy. Low-molecular-weight samples of relatively broad molecular weight distribution at low yields were obtained. The linear polymer was further functionalized upon reaction with potassium ethyl xanthate, PEX, to afford the corresponding macro-CTA. The reaction took place in acetone solution at room temperature. NMR and IR spectra revealed a quantitative incorporation of the CTA moieties along the polymer backbone. Subsequent RAFT polymerization of NVP yielded the desired graft copolymers applying the grafting “from” methodology and the R-group approach (Figure 92). The reaction was conducted in anisole solution at 60 °C, employing AIBN as initiator. Unfortunately, SEC characterization was not provided. Therefore, conclusions regarding the molecular weight distribution of the brushes along with the possible appearance of side reactions during the synthesis cannot be given. These bottle-brushes were found to self-assemble into micelles in aqueous solutions. The model hydrophobic compound Nile red was efficiently encapsulated into these micelles.

Bottle-brushes consisting of a polymethacrylate backbone along with PLLA and PNVP side chains were prepared by RAFT and ROP techniques utilizing the grafting “from” methodology and the R-group approach, following the reaction series given in Figure 93 [182]. The backbone was synthesized via the RAFT polymerization of 2-[(2-bromopropanoyl)oxy]ethyl methacrylate, BPEM, in toluene at 65 °C employing AIBN and either cumyl dithiobenzoate, CDB, or dicumyl tetrathioterephthalate, DCTP, as the CTA. DCTP is a difunctional CTA allowing the growth of the polymer chain to both directions. Subsequent addition of hydroxypropyl methacrylate, HPMA, afforded the corresponding diblock and triblock copolymers, PBPEM-b-PHPMA and PHPMA-b-PBPEM-b-PHPMA, respectively. The polymerization was conducted in N,N-dimethyl acetamide at 65 °C in the presence of the radical initiator AIBN. The pendant bromine groups of these diblock and triblock copolymers were converted to CTA moieties via reaction with sodium diphenyldithiocarbamate. The transformation took place in THF solution at room temperature. SEC and NMR analysis revealed that the backbones had relatively low molecular weights and low dispersity values (Ð ≤ 1.20). The functionalization reaction caused no damage to the backbone (degradation, crosslinking, etc.) and it was quantitative. Therefore, linear chains were obtained carrying both pendant hydroxyl groups, capable of initiating ROP of LLA and CTA functions, and promoting the polymerization of NVP, leading to the synthesis of bottle-brushes with both PLLA and PNVP side chains. The ROP of LLA was conducted first in N,N-dimethyl acetamide at 30 °C in the presence of the organocatalyst 1,8-diazabicyclo[5,40]undec-7-ene, DBU. SEC analysis confirmed the presence of the desired brush along with linear PLLA. The linear byproduct was eliminated by fractionation. The RAFT of NVP was realized in a second step again in N,N-dimethyl acetamide at 60 °C, using AIBN as the radical source. Tailing effects in SEC traces were attributed to the presence of linear PNVP chains, due to radical transfer reactions to the monomer. Very high grafting density can be achieved through this procedure. However, the characterization data were not enough to exclude the possibility of initiation sites that remained unreacted either at the ROP or RAFT step of polymerization.

A different approach was applied for the synthesis of PVDF-g-PNVP graft copolymers, where PVDF is poly(vinylidene fluoride) [183]. Peroxide initiation sites were immobilized on the PVDF chains, upon treatment of PVDF solutions in DMF with a continuous stream of O_3_/O_2_ at room temperature. A peroxide content of 10^−4^ mol.g^−1^ was achieved in 15 min. Subsequent addition of NVP promoted the RAFT polymerization of the monomer in DMF solutions at 60 °C in the presence of 1-phenylethyl dithiobenzoate, as the CTA (Figure 94). The PNVP chains were further extended with the addition of DMAEMA. The polymerization was conducted in 2-propanol at 70 °C employing AIBN as the initiator, thus producing the PVDF-g-(PNVP-b-PDMAEMA) graft terpolymers. The grafting process was confirmed by NMR and FT-IR spectroscopies. However, molecular characterization data are totally missing, and thus comments regarding control of the polymerization, possible presence of impurities, grafting density, etc., cannot be made. The antifouling and antibacterial properties of membranes prepared from these graft copolymers were thoroughly examined.

## 5. Conclusions

Recent advances in the controlled RAFT polymerization of NVP have allowed the synthesis of complex macromolecular architectures based on poly(N-vinyl pyrrolidone), PNVP, including statistical, block, star and graft copolymers. RAFT polymerization techniques and combinations with other controlled/living methodologies have been successfully employed. Special interest should be given to the fact that NVP belongs to the less activated monomers, thus making it tricky to be combined with the more activated monomers. In addition, the presence of several termination reactions in the RAFT process leads to the formation of byproducts, especially when multistep reaction sequences are employed. In order to avoid these situations, kinetic control of the experiments is required, along with special care in purification of the starting materials (monomers, solvents, CTAs, initiators, etc.) and the final polymeric products. Careful selection of the CTAs, the solvent and the sequence of monomer addition may play an important role in matching the reactivities of the various monomers and thus leading to the synthesis of well-defined products. In addition, the presence of specific end groups of the polymer chains coming from the suitable CTA may allow the combination of RAFT with other polymerization techniques, such as ATRP and ROP. Upon increasing the complexity of the desired structure to be prepared, the more difficult the effort becomes. Towards this direction, click chemistry offers a valuable tool for the synthesis of complex macromolecular architectures. In any case, advanced characterization and purification techniques have to be applied. Under these conditions. well defined structures can be obtained with very good control over the molecular characteristics. These results will definitely lead to novel applications in many different scientific sectors.

## Data Availability

Not applicable.

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
