# Peer review of "Recent Advances in the Synthesis of Complex Macromolecular Architectures Based on Poly(N-vinyl pyrrolidone) and the RAFT Polymerization Technique"

_polymers, 2022, doi:10.3390/polym14040701_

Round 1

Reviewer 1 Report

Recent advances in the controlled RAFT polymerization of complex macromolecular architectures based on poly(N-vinyl pyrrolidone), PNVP, are summarized in this review article. The paper will be interesting for researchers working in polymer field. The paper could be accepted after revision.

-I would recommend to change title of the paper. The title should show that this is the review article.

-2 fields are mentioned as b)  in lines 34 and 36

- I would not recommend to describe the method of Reversible Addition-Fragmentation chain Transfer (RAFT) polymerization in details. This is well known information in polymers field ?

-Preparation of the polymers/copolymers by using other polymerization methods should be also mentioned in the review and properties of the obtained materials should be compared.

-The authors should provide some recommendations in conclusions of the paper.  After the review they are able to give some suggestions for synthesis and application of the polymers/copolymers.

Author Response

Reviewer 1

We are grateful to the reviewers for their fruitful comments and suggestions. We have taken everything into consideration and revised our manuscript accordingly. All changes in the text are given in red color. Our answers point by point to the reviewers are given below:

Recent advances in the controlled RAFT polymerization of complex macromolecular architectures based on poly(N-vinyl pyrrolidone), PNVP, are summarized in this review article. The paper will be interesting for researchers working in polymer field. The paper could be accepted after revision.

-I would recommend to change title of the paper. The title should show that this is the review article.

We agree with the reviewer to revise the title of the manuscript. The proposed new title is the following: “Recent advances in the synthesis of complex macromolecular architectures based on poly(N-vinyl pyrrolidone) and the RAFT polymerization technique. A comprehensive review”

-2 fields are mentioned as b)  in lines 34 and 36

We are sorry for the mistake. We have corrected the text.

- I would not recommend to describe the method of Reversible Addition-Fragmentation chain Transfer (RAFT) polymerization in details. This is well known information in polymers field ?

RAFT is the basic polymerization technique mentioned in this review article. We believe that it is important to mention the basics of this methodology for two major reasons: a) the article will be available to people who are not familiar with this technique and essential information is needed and b) in order to have a complete picture of the mechanistic considerations involved in the polymerization of NVP and how this is connected with the polymerization with either less activated or more activated monomers. The method is described in less than two pages of the review article.

-Preparation of the polymers/copolymers by using other polymerization methods should be also mentioned in the review and properties of the obtained materials should be compared.

We focus on the RAFT polymerization technique because it is the only method providing control over the polymerization of NVP. It is well known that NVP can be only polymerized via radical polymerization. Other controlled radical polymerization techniques, such as Atom Transfer (ATRP) and Nitroxide Mediated (NMP) Radical polymerization techniques have not been proven versatile methodologies to provide complex macromolecular architectures of PNVP. Discussion on the general procedures applied for the synthesis of various architectures is given in the text and connection with PNVP structures is provided. However, a thorough report on the synthesis of complex macromolecular architectures is a huge task and beyond the scope of this review article. For the same reason we decided to avoid detailed analysis on the applications of these PNVP structures. Our focus and main interest is the synthesis of these polymers. However, in the Introduction we included a general report on the applications of PNVP based materials and within the text we have scattered reports on the applications of the reported materials. According to the suggestion of the Reviewer we have added more information on this issue.

-The authors should provide some recommendations in conclusions of the paper.  After the review they are able to give some suggestions for synthesis and application of the polymers/copolymers.

We agree with the Reviewer and we have added the requested information in the conclusion section of this manuscript.

Reviewer 2 Report

The manuscript “Synthesis of complex macromolecular architectures based on 2 poly(N-vinyl pyrrolidone) and the RAFT polymerization technique” by Roka et al. provides a detailed review of the RAFT polymerization of NVP in various molecular structures such as block copolymer, star & graft copolymers, and have discussed the reactivity of NVP with other types of monomers. Overall the manuscript is well written for its technical part, but the reviewer considers some additional polishing is required, as the connection is not well-established between the current discussion to a comprehensive review with a good storyline. I have a few comments as below:

  1. For example, in the introduction, though the history aspect and potential applications of the NVP, or PNVP was briefly discussed, but the applications were only listed instead of explaining their correlation with the controlled polymerization, and why the reaction should be controlled, especially of using the RAFT, rather than other polymerization techniques. In other words, it is challenging for readers to catch ‘complex macromolecular architecture’ to the requirements in the application, and why that certain chemistry is in demand. The reviewer would urge authors to provide a detailed review of applications and highlight why the certain molecular shape is beneficial to certain applications. The reviewer feels the current manuscript is more like a book section instead of a review paper.

  1. The same applies to the RAFT, in which the mechanism was discussed in detail but the application aspect of using the RAFT was not reviewed in detail. Perhaps some comment over the preference on RAFT (and the associated reason why using RAFT over different structures) is needed here.

  1. It seems that the physical characteristics of the complex molecular architecture associated with the corresponding performance were rarely discussed, and hence it was hard to capture why certain molecular geometry (or copolymer composition) is of interest at the beginning of each section. It would be nice to mention if certain copolymers were synthesized to target a specific application or tackle a synthetic challenge to make the review with a better storyline.

  1. Some molecular structures listed in the manuscript are hard to follow, and sometimes inconsistent in their format. The reviewer suggests using the ACS format for chemical structures.

  1. In some sections e.g., 4.2.4, the discussion was listed as bullet points, similar to the format of a presentation instead of a comprehensive review paper. The reviewer suggests listing these information in a table with the appropriately labeled reference numbers.

Author Response

Reviewer 2

We are grateful to the reviewers for their fruitful comments and suggestions. We have taken everything into consideration and revised our manuscript accordingly. All changes in the text are given in red color. Our answers point by point to the reviewers are given below:

The manuscript “Synthesis of complex macromolecular architectures based on poly(N-vinyl pyrrolidone) and the RAFT polymerization technique” by Roka et al. provides a detailed review of the RAFT polymerization of NVP in various molecular structures such as block copolymer, star & graft copolymers, and have discussed the reactivity of NVP with other types of monomers. Overall the manuscript is well written for its technical part, but the reviewer considers some additional polishing is required, as the connection is not well-established between the current discussion to a comprehensive review with a good storyline. I have a few comments as below:

  1. For example, in the introduction, though the history aspect and potential applications of the NVP, or PNVP was briefly discussed, but the applications were only listed instead of explaining their correlation with the controlled polymerization, and why the reaction should be controlled, especially of using the RAFT, rather than other polymerization techniques. In other words, it is challenging for readers to catch ‘complex macromolecular architecture’ to the requirements in the application, and why that certain chemistry is in demand. The reviewer would urge authors to provide a detailed review of applications and highlight why the certain molecular shape is beneficial to certain applications. The reviewer feels the current manuscript is more like a book section instead of a review paper.

 The detailed report of applications of the PNVP based structures is outside the scope of this review article. We focus on the synthesis of complex macromolecular architectures of PNVP. This issue has never been examined before the rise of the RAFT polymerization technique. It is well known that NVP can be polymerized only through radical polymerization. The other controlled radical polymerization techniques (Atom Transfer, ATRP and Nitroxide Mediated, NMP, radical polymerization) have been proven inappropriate to provide well-controlled macromolecular structures of PNVP. Therefore, this area of polymer synthesis is still unexplored and especially the synthesis of structures more complex than linear block copolymers. This manuscript aims to cover this gap in the literature providing a comprehensive review of the possibilities we have to synthesize complex structures based on PNVP. Not all of the examples provided in the literature are well-defined products. Our main purpose was to discuss the synthetic challenges and opportunities in this new field of Polymer Synthesis. It is well known that the macromolecular architecture affects the properties of the polymeric materials. It is outside the scope of this work to describe in detail the solution and solid state properties of the PNVP based materials and their applications. However, a general discussion on the applications of PNVP polymers is given in the Introduction and more data were added within the text. We also tried to clarify these issues, raised by the Reviewer, in the text.

  1. The same applies to the RAFT, in which the mechanism was discussed in detail but the application aspect of using the RAFT was not reviewed in detail. Perhaps some comment over the preference on RAFT (and the associated reason why using RAFT over different structures) is needed here.

 As was mentioned previously RAFT is the only technique, which is able to provide well-defined complex macromolecular architectures based on PNVP. This is the reason why we focus on this methodology. This point was further clarified in the text.

  1. It seems that the physical characteristics of the complex molecular architecture associated with the corresponding performance were rarely discussed, and hence it was hard to capture why certain molecular geometry (or copolymer composition) is of interest at the beginning of each section. It would be nice to mention if certain copolymers were synthesized to target a specific application or tackle a synthetic challenge to make the review with a better storyline.

The detailed discussion of properties and applications of these materials is outside the scope of this review article. However, as the Reviewer suggested, more information regarding the applications of the PNVP based complex polymeric structures were added in the text.

  1. Some molecular structures listed in the manuscript are hard to follow, and sometimes inconsistent in their format. The reviewer suggests using the ACS format for chemical structures.

 We tried to use the format that was given in the original articles helping the reader to have a direct connection with the initial work of the various authors. However, when the situation was confusing and in order to have a more homogeneous presentation we changed the way of presentation of the molecular structures.

  1. In some sections e.g., 4.2.4, the discussion was listed as bullet points, similar to the format of a presentation instead of a comprehensive review paper. The reviewer suggests listing these information in a table with the appropriately labeled reference numbers.

We do not usually employ bullets within the text, since we tried to explain every point described in the manuscript. The specific mention of the Reviewer refers to the synthesis of triblocks and specifically to the various types of these kinds of materials (symmetric and asymmetric triblock copolymers, along with triblock terpolymers). This is well known in the literature and this issue has been frequently described in review papers and books (including the RAFT methodology). The synthesis of all the triblocks containing PNVP blocks was reported and discussed in detail within the text. 

Round 2

Reviewer 1 Report

If editor and other reviewers agree I would also recommend the paper for publication after the revision.

Reviewer 2 Report

Changes are made to comments and I would recommend publication.